# Pan-cancer analysis of longitudinal metastatic tumors reveals genomic alterations and immune landscape dynamics associated with pembrolizumab sensitivity

S. Y. Cindy Yang[1], Scott C. Lien[2], Ben X. Wang[3], Derek L. Clouthier[3], Youstina Hanna[3], Iulia Cirlan[3], Kelsey Zhu[3], Jeffrey P. Bruce[3], Samah El Ghamrasni [3], Marco A. J. Iafolla[3,4], Marc Oliva [3,4], Aaron R. Hansen[3,4], Anna Spreafico[3,4], Philippe L. Bedard [3,4], Stephanie Lheureux[3,4], Albiruni Razak[3,4], Vanessa Speers[3], Hal K. Berman [3], Alexey Aleshin[5], Benjamin Haibe-Kains[1,3,6,7,8], David G. Brooks[2,3], Tracy L. McGaha [2,3], Marcus O. Butler [2,3,4], Scott V. Bratman [1,3,9], Pamela S. Ohashi[2,3], Lillian L. Siu [3,4,10]✉ & Trevor J. Pugh [1,3,6,10]✉

Serial circulating tumor DNA (ctDNA) monitoring is emerging as a non-invasive strategy to predict and monitor immune checkpoint blockade (ICB) therapeutic efficacy across cancer types. Yet, limited data exist to show the relationship between ctDNA dynamics and tumor genome and immune microenvironment in patients receiving ICB. Here, we present an in-depth analysis of clinical, whole-exome, transcriptome, and ctDNA profiles of 73 patients with advanced solid tumors, across 30 cancer types, from a phase II basket clinical trial of pembrolizumab (NCT02644369) and report changes in genomic and immune landscapes (primary outcomes). Patients stratified by ctDNA and tumor burden dynamics correspond with survival and clinical benefit. High mutation burden, high expression of immune signatures, and mutations in *BRCA2* are associated with pembrolizumab molecular sensitivity, while abundant copy-number alterations and *B2M* loss-of-heterozygosity corresponded with resistance. Upon treatment, induction of genes expressed by T cell, B cell, and myeloid cell populations are consistent with sensitivity and resistance. We identified the upregulated expression of *PLA2G2D*, an immune-regulating phospholipase, as a potential biomarker of adaptive resistance to ICB. Together, these findings provide insights into the diversity of immunogenomic mechanisms that underpin pembrolizumab outcomes.

[1] Department of Medical Biophysics, University of Toronto, Toronto, ON, Canada. [2] Department of Immunology, University of Toronto, Toronto, ON, Canada. [3] Princess Margaret Cancer Centre, University Health Network, Toronto, ON, Canada. [4] Division of Medical Oncology & Haematology, Princess Margaret Cancer Centre, University of Health Network, Department of Medicine, University of Toronto, Toronto, ON, Canada. [5] Natera, Inc., San Carlos, CA, USA. [6] Ontario Institute for Cancer Research, Toronto, ON, Canada. [7] Department of Computer Science, University of Toronto, Toronto, ON, Canada. [8] Vector Institute, Toronto, ON, Canada. [9] Department of Radiation Oncology, University of Toronto, Toronto, ON, Canada. [10] These authors jointly supervised this work: Lillian L. Siu, Trevor J. Pugh. ✉email: lillian.siu@uhn.ca; trevor.pugh@utoronto.ca

Therapeutic blockade of the programmed cell death 1 (PD-1) immune checkpoint has provided durable clinical benefit to patients with advanced cancers. When PD-1 receptor signaling is abrogated, effector functions of tumor-specific CD8 + T lymphocytes within the tumor microenvironment (TME) can be restored, resulting in disease control. While high clinical benefit rates (37–87%) have been observed in Hodgkin's lymphoma, metastatic melanoma, Merkel cell carcinoma, and microsatellite unstable cancers[1], modest response rates (10–20%)[1] coupled with the development of immune-related adverse events in a fraction of patients, have continued to drive research into strategies to improve patient selection for ICB therapy[2].

High tumor mutation burden (high-TMB) has emerged as the most promising and controversial pan-cancer biomarker for predicting ICB therapeutic responses[3,4]. Despite pan-cancer US FDA approval of ICB treatment for any high-TMB tumor, high-TMB status failed to predict improved ICB response across cancer types in a recent assessment with over 1500 tumors[4], calling into question its clinical utility. In recent years, tumor genomics studies enabled by large, multi-dimensional datasets—such as The Cancer Genome Atlas (TCGA), identified links between genomic alterations in cancers, infiltrating immune cell populations and spontaneous local immune cytolytic activity[5] to suggest that immune evasion strategies are fundamental to tumor development and may impact ICB response across cancer types. Molecular characteristics uncovered in these studies have potential as predictive biomarkers or therapeutic targets to improve ICB clinical benefit. Together, this forms a strong rationale to pursue ICB response biomarker discovery and assessments in pan-cancer cohorts.

We previously reported that the change in circulating tumor DNA (ΔctDNA) level at 6–7 weeks of pembrolizumab treatment from baseline ctDNA correlated with progression-free survival (PFS), overall survival (OS), and clinical benefit (CB)[6] in metastatic cancer patients enrolled in the INvestigator-initiated Phase-II Study of Pembrolizumab Immunological Response Evaluation clinical trial (INSPIRE; NCT02644369). To extend these findings, we explored the use of ΔctDNA as an indicator of molecular sensitivity to stratify patients into subgroups with or without robust response to pembrolizumab for downstream comparisons of tumor genomic and gene-expression features. Evaluating changes in genome and immune biomarkers in patient blood and tumor biopsies is the primary outcome for the INSPIRE trial. This analysis draws on the clinical and molecular profiling data from all 106 patients enrolled, building upon a previous interim report[7] of the first 80 patients and germline HLA status from 101 patients[8] (Supplementary Fig.1).

Here, we show that combined use of ΔctDNA and tumor burden change define patient populations with distinct survival outcomes. We also provide evidence that this combination biomarker may differentiate tumor pseudoprogression from true progression. Based on this classification, we perform an in-depth integrated analysis of genomic and transcriptomic data derived from tumor biopsies collected from the INSPIRE trial and report the primary outcomes by comparing somatic mutations, copy number alterations (SCNAs), tumor immune microenvironment, and gene-expression patterns between selected patients with the lowest level of molecular responses and those with high response or clinical benefit. Comparisons of baseline somatic and germline mutation and copy-number profiles between sensitivity groups reveals frequently altered genes associated with early molecular response to pembrolizumab. Gene-expression profiling of longitudinal tumor tissue samples identifies differentially regulated genes associated with clinical benefit and disease progression. This study demonstrates the potential of ΔctDNA-guided sample stratification for biomarker assessment and discovery. Furthermore, we highlight the added value of longitudinal tumor and blood specimen integrated molecular profiling to uncover diverse mechanisms of response and resistance to ICB.

## Results

**Early changes in ctDNA and tumor burden defines pembrolizumab sensitivity phenotypes associated with clinical outcomes and mutation burden.** In the INSPIRE cohort ($n = 106$) of advanced solid cancer patients treated with pembrolizumab (head and neck (HNSCC), triple-negative breast (TNBC), high-grade serous carcinoma (HGSC), melanoma (MM), and other mixed solid tumors (MST)), we observed 4% ($n = 4$) complete response (CR), 13% ($n = 14$) partial response (PR), 25% ($n = 27$) stable disease (SD), 54% ($n = 57$) progressive disease (PD) (Fig. 1A) as per RECIST criteria. A quarter of the patients ($n = 26$) achieved CB, defined as CR, PR, and SD longer than 18 weeks (Supplementary Data 1). At the date of data collection cut-off (18 July 2019), the median follow-up duration was 11 months (range 0.6–35 months).

We performed whole-exome sequencing (WES) of baseline tumor biopsies with matching peripheral blood derived germline DNA to identify somatic alterations, of which one-third ($n = 34$) of the samples had low DNA content or quality unsuitable for WES. Of the 72 patients with successfully profiled baseline tumors, 71 had undergone longitudinal assessment of ctDNA dynamics[6] and tumor burden assessment by imaging at matching time points (Fig. 1B and Supplementary Fig.1).

When assessing the correlation between the change of ctDNA (ΔctDNA) at 6–7 weeks and the change of target lesion measurement (ΔTM) between baseline and 9 weeks of treatment, we observed that the combination of these metrics stratified patients into four subgroups associated with distinct survival outcomes (Fig. 1C–E). As such, we defined this classification system as a proxy of molecular sensitivity to pembrolizumab. Our cohort included: 44% (32/71) low sensitivity (LS, ΔctDNA, and ΔTM positive), 22% (16/71) high sensitivity (HS, ΔctDNA, and ΔTM negative), 22% (16/71) mixed sensitivity with potential pseudoprogression (MSPP, ΔctDNA negative, and ΔTM positive), and 10%(7/71) mixed sensitivity with emerging resistance (MSER, ΔctDNA positive, and ΔTM negative). HS and MSPP groups had improved OS and PFS compared to LS and MSER (Fig. 1D, E), while the HS group had the most significant survival improvement compared to LS (OS HR = 0.54, log-rank $p < 0.001$; PFS HR = 0.048, log-rank $p < 0.001$). MSPP patients experienced longer OS despite shorter PFS as compared to MSER, suggesting that while both MSPP and MSER groups are characterized by mixed responses, some MSPP patients may have experienced pseudoprogression (an increase in tumor size as a result of increased immune cell content within the tumor tissue), previously indistinguishable from true progression by tumor imaging alone. All patients within the LS cohort had disease progression while the HS group had the highest CB rate (88%, 14/16) and the largest changes in TM and ctDNA (Fig. 1C). Both HS patients without CB (a TNBC and a HNSCC) survived beyond 2 years (within the top 25% OS of all patients treated) after stopping treatment.

To determine the concordance between pembrolizumab sensitivity subgroups and established ICB predictive biomarkers based on pre-specified universal cutoffs, we evaluated the distributions of TMB and PD-L1 protein expression within the HS subgroup. TMB as a continuous measure was significantly higher in HS tumors compared to the overall cohort (mean 7.72 mut/Mb vs. 1.74 mut/Mb, $p < 0.05$, one-sided Kolmogorov–Smirnov test) (Fig. 1F). The majority (14 of 16) of HS TMB fell within the top-tertile of the

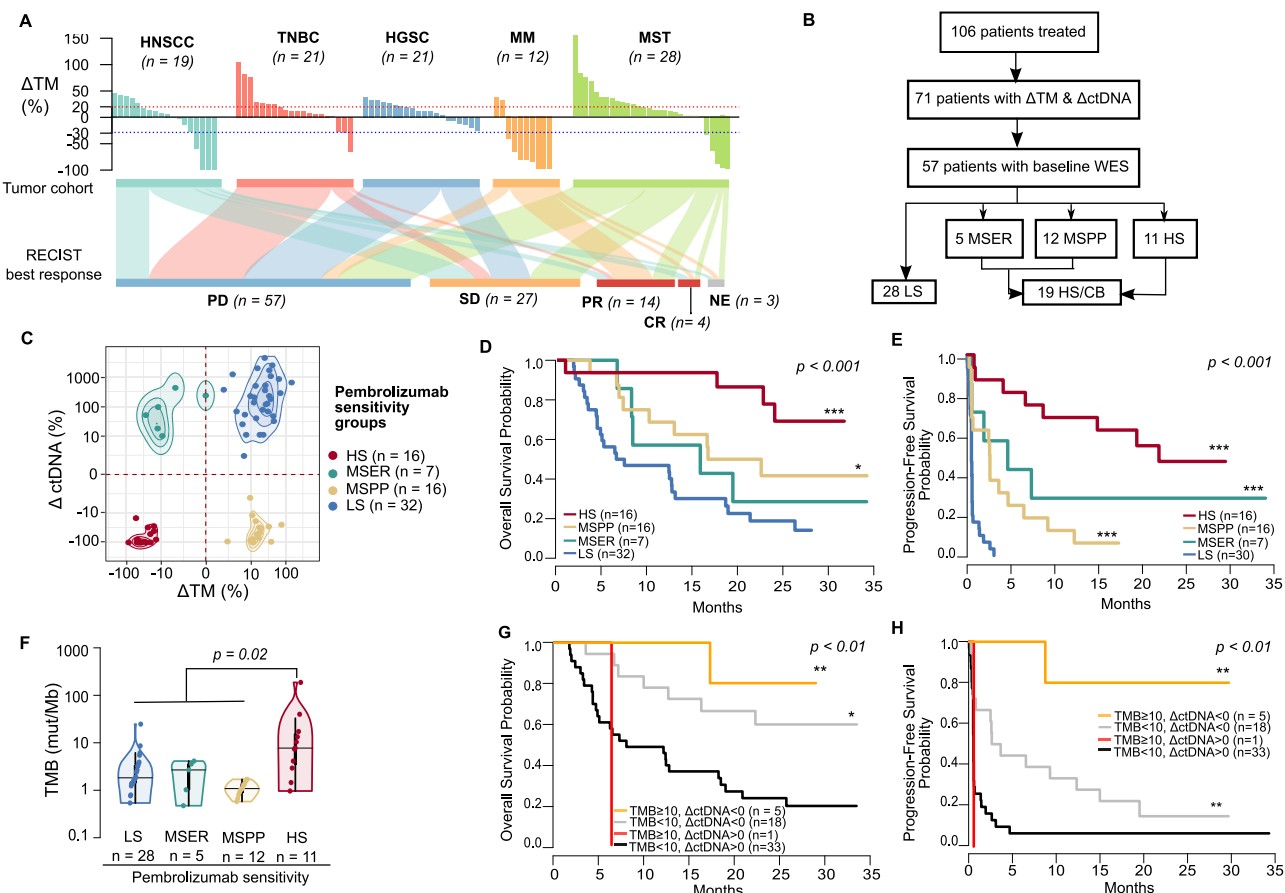

**Fig. 1 INSPIRE patient stratification by early ΔctDNA and ΔTM. A** Overview of change of tumor measurement (ΔTM) at best response of all patients enrolled in the INSPIRE clinical trial ($n = 106$). Patients are grouped by cancer cohort and sorted in order of decreasing ΔTM. Below the waterfall plot, widths of joining lines reflect the number of patients within the corresponding tumor cohort and best overall response by RECIST. **B** Simplified consort diagram summarizing the number of patients with complete ΔTM, ΔctDNA, and baseline WES profiling stratified into the four pembrolizumab sensitivity groups. LS low-sensitivity, MSER mixed-sensitivity with emerging resistance, MSPP mixed-sensitivity with potential pseudoprogression, HS high-sensitivity. **C** Distribution of ΔTM and ΔctDNA for each patient. Contours around each group indicate the density within each group with the median shown as the center of the contour. Points are colored according to pembrolizumab sensitivity groups. Dotted red vertical and horizontal lines indicate ΔTM = 0 and ΔctDNA = 0. **D** Kaplan–Meier plot of overall survival from treatment cycle 3 in patients grouped by pembrolizumab sensitivity. ($p = 0.00025$). **E** Kaplan–Meier plot of progression-free survival from treatment cycle 3 in patients grouped by pembrolizumab sensitivity. ($p = 8.7 \times 10^{-11}$) **F** Comparison of TMB distribution between pembrolizumab sensitivity groups. The median for each group is shown as a horizontal line and data points are shown ordered by increasing value. The distance between the third-quartile (Q3) and first-quartile (Q1), known as the interquartile range (IQR), is marked around the median by a black rectangle. Vertical lines extending from the top and bottom of the rectangle show the maximum (Q3 + 1.5-times IQR) and minimum (Q1 + 1.5-times IQR). *P*-values shown are calculated by a two-sided Kolmogorov–Smirnov test. **F** Kaplan–Meier plot for overall survival from treatment cycle 3 in patients stratified by TMB ≥ 10 and ΔctDNA. ($p = 0.0042$). **G** Kaplan–Meier plot for overall survival from treatment cycle 3 in patients stratified by TMB ≥ 10 and ΔctDNA. ($p = 0.0053$). *P*-values for Kaplan–Meier plots reflect the outcomes of log-rank tests for the survival model. **H** Kaplan–Meier plot for progression-free survival from treatment cycle 3 in patients stratified by TMB ≥ 10 and ΔctDNA. ($p = 0.00018$). *P*-values for Kaplan–Meier plots reflect the outcomes of log-rank tests for the survival model. Statistical significance for all panels: \*\*\*$p \leq 0.001$; \*\*$p \leq 0.01$; \*$p \leq 0.05$; +$p \leq 0.10$. Source data are provided in SourceData_Fig1.xlsx.

study distribution, of which only 6 tumors (6 of 16; 38%) met the TMB-high biomarker criteria (TMB ≥ 10 mut/Mb). PD-L1 expression was only detected (PD-L1 MPS > 50%) in half (5 of 10) of the HS tumors at baseline with available immunohistochemistry (IHC) data (Supplementary Fig. 2). Together, the discordance between TMB-high, PD-L1, and HS classification, suggest that treatment opportunities in otherwise treatment-sensitive patients would be lost when using current pre-specified cutoffs of TMB and PD-L1 scoring to select patients for anti-PD1 treatment in a pan-cancer setting.

Given that TMB-high (TMB ≥ 10 mut/Mb) status is a US FDA-approved pan-cancer biomarker used to select metastatic cancers

for ICB therapy, we explored whether ΔctDNA provides added benefit for risk stratification within TMB-high and TMB-low tumors (TMB < 10 mut/Mb). We compared the OS and PFS of TMB-high and TMB-low groups further divided based on ΔctDNA (Fig. 1G, H). As expected compared to the patients with the least favorable responses (TMB-low and ΔctDNA > 0, $n = 33$), we observed the most favorable OS and PFS probabilities in TMB-high and ΔctDNA < 0 (OS HR = 0.32, $p = 0.007$; PFS HR = 0.41, $p = 0.008$, Cox proportional hazards), followed by TMB-low and ΔctDNA < 0 ($n = 18$) (OS HR = 0.14, $p = 0.05$; PFS HR = 0.05, $p = 0.005$, Cox proportional hazards). These data suggest that ΔctDNA status provides added value to predict

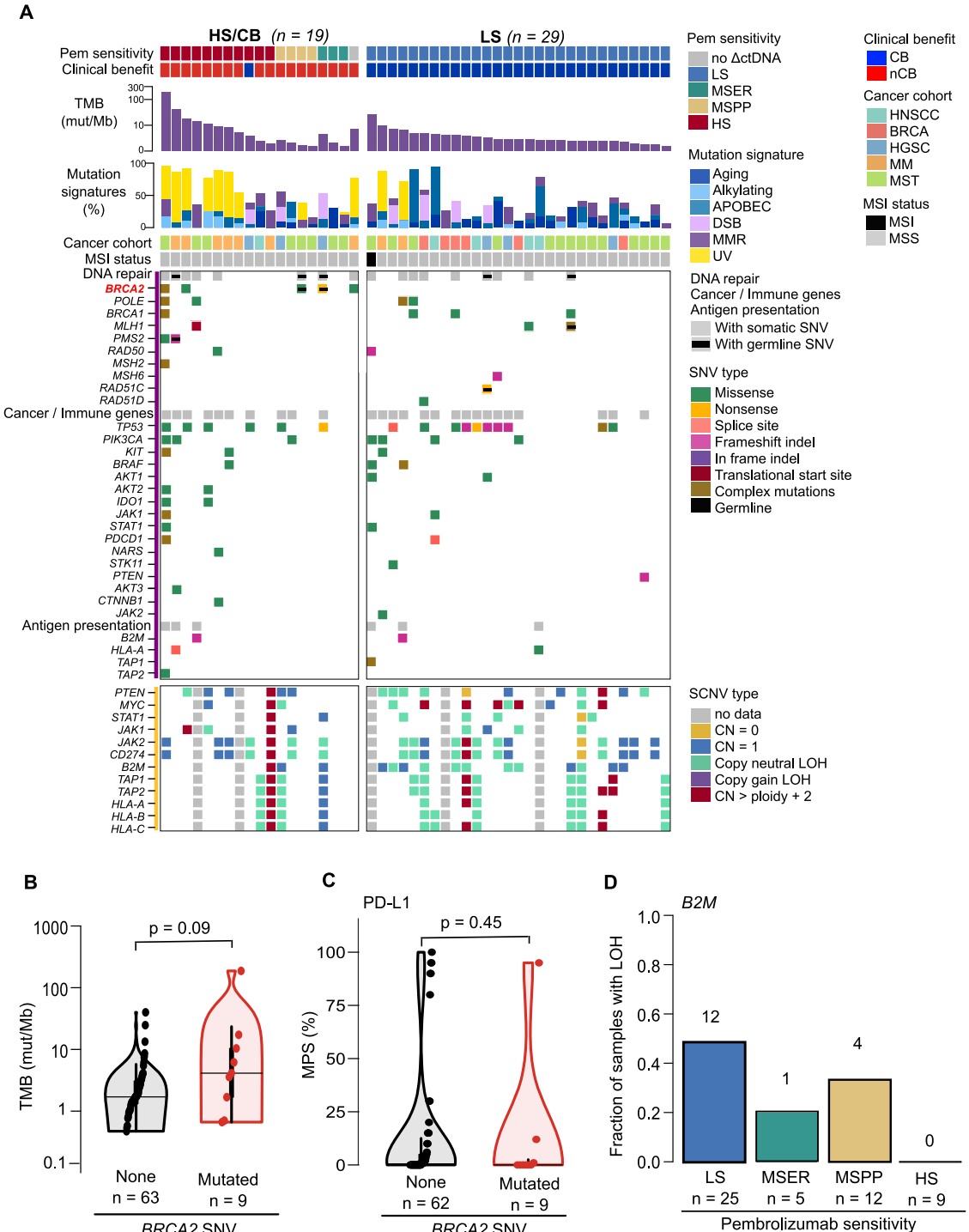

pembrolizumab response within predetermined TMB-high and TMB-low subgroups.

**BRCA2 is frequently altered in baseline tumors from pembrolizumab-treated patients with clinical benefit.** To uncover genomic alterations associated with pembrolizumab response, we identified genes that were more frequently altered by non-synonymous germline or somatic mutations in patients who experienced CB or high pembrolizumab sensitivity (HS/CB; $n = 19$) compared to patients with low sensitivity (LS; $n = 27$). We identified 35 genes that were more frequently altered in HS/CB tumors versus corresponding TCGA cohorts; of which, 8

genes (*PREX2, BIRC6, CNTNAP2, PTPRB, GRM3, ANKRD11, BRCA2,* and *TET3*) are listed on the OncoKB Cancer Genes catalogue (Supplementary Data 2). *BRCA2* was mutated in 5 HS/CB (26%) and none in LS of mixed cancer types (TNBC, Merkel cell carcinoma, HGSC, MM, sarcoma, and basal cell carcinoma) (Fig. 2A). *BRCA2* mutation frequency was 12.5% (9 of 72) in our data set and similarly distributed across cancer cohorts (HGSC: 0/19, TNBC: 1/21, HGSC 2/21, MM: 2/12, and Mixed: 4/28). Increased frequency of *BRCA2* mutations was also identified from a similar analysis reported by Hugo et al.[9] in a cohort of meta-static melanoma patients. Interestingly, mutations in *BRCA1* were not enriched in either group (HS/CB = 1/19, LS = 3/27 (Fig. 2A). No single gene was found to be significantly enriched for

**Fig. 2 Somatic alterations in patients sensitive and resistant to pembrolizumab. A** Co-mutation plot of mutation burden, mutation signatures, somatic, and germline mutations (purple, left) in selected DNA repair genes, known cancer and immune evasion related genes, and antigen presentation genes. Copy number alterations for selected immune evasion and antigen presentation genes are also shown (yellow, left). Samples are grouped as pembrolizumab-sensitive (in the HS/CB group) and resistant (in the LS group) and sorted in order of decreasing TMB within each group. Genes are grouped by curated categories and shown in order of decreasing frequency across both groups. *BRCA2* (bold) had significantly higher mutation frequency in the ICB sensitive group. For the complete list of curated genes, refer to Supplementary Table 1. Only genes with alterations are shown. **B** Comparison of TMB between samples with or without *BRCA2* mutations. **C** Comparison of PD-L1 expression by IHC staining between samples with or without *BRCA2* mutations. For violin plots in **B** and **C**, the median for each group is marked by a horizontal black line in the center of a rectangle and data points are shown as open circles. The distance between the third-quartile (Q3) and first-quartile (Q1), known as the interquartile range (IQR), is marked around the median by a rectangle. Vertical lines extending from the top and bottom of the rectangle show the maximum (Q3 + 1.5-times IQR) and minimum(Q1 + 1.5-times IQR). Statistical significance was determined using two-sided Wilcoxon rank-sum tests. **D** Comparison of *B2M* LOH frequency in pembrolizumab sensitivity groups. The number of samples with *B2M* LOH within each sensitivity group is annotated above each bar. Source data are provided in SourceData_Fig2.xlsx. LS low sensitivity, HS/CB high-sensitivity/clinical benefit, TMB tumor mutation burden, ICB immune checkpoint blocker, IHC immunohistochemistry, LOH loss of heterozygocity.

---

mutations in the LS group, possibly due to the large number of alternate and overlapping signaling transduction pathways that can manifest immune evasion and escape phenotypes.

We further investigated the relationship of *BRCA2* mutation status with known biomarkers of response: TMB and PD-L1 protein expression. We observed that tumors with germline and/or somatic *BRCA2* mutation have significantly higher TMB compared to tumors without mutated *BRCA2* ($p < 0.10$, Wilcoxon rank-sum test, two-sided) (Fig. 2B), consistent with previous studies[9]. We further validated this significant association using publicly available mutation and TMB data from three independent publicly available pan-cancer data sets (Broad MSS mixed solid tumors[10], UMich MET500[11], and MSKCC-IMPACT IO study[12]) (Supplementary Fig. 3). While we observe a higher clinical benefit rate (CBR = 35%, 18 of 51) in PD-L1 expressing tumors (PD-L1 MPS > 0%) compared to tumors without PD-L1 expression (CBR = 15%, 8 of 52), PD-L1 was only expressed (MPS > 0%) in 3 of 9 tumors with *BRCA2* mutations (Fig. 2C). This finding suggests that *BRCA2* mutation status may predict pembrolizumab response, independent of PD-L1 expression.

**High percent genome copy number alterations and *B2M* LOH are associated with intrinsic resistance to pembrolizumab**. We sought to describe the prevalence of immune evasion through somatic genomic alterations in our cohort by assessing the frequency of somatic single-nucleotide variants (SNV) within a compendium of 49 genes associated with resistance to immune checkpoint blockade or immune-suppressed TME (Supplementary Table 1 and Fig. 2A)[13,14]. At least one gene in the catalog was altered in 68% (49/72) of all sequenced baseline tumors; with *TP53* being the most frequently mutated (48.6%, 35/72). Mutations in the genes associated with antigen presentation (*HLA-A, HLA-B, HLA-C, TAP1, TAP2,* and *B2M*) and interferon-gamma pathways (*STAT1, STAT3, JAK1, JAK2, IFNGR1,* and *IFNGR2*) were identified at a low frequency (8%, 6/72). Loss of heterozygosity (LOH) of at least one of three HLA class I loci was observed in 39% (24/61, 11 patients had at least one homozygous HLA class I locus) while somatic mutation occurred in only 0.5% (4/72) of tumors prior to anti-PD-1 treatment. Together, this data provides evidence that genomic alterations associated with immune evasion are diverse and prevalent in solid tumors.

We previously reported that a high baseline percent genome with copy number alterations (PGA) is significantly associated with progressive disease[7]. With the updated data set, we confirmed this observation by comparing baseline PGA between HS/CB and LS ($p = 0.01$, Wilcoxon rank-sum test, two-sided, Supplementary Fig. 4). These results show that PGA-high status (upper cohort tertile) is associated with inferior OS and PFS compared to the remaining cohort (Supplementary Fig. 5). When

combined with TMB to predict pembrolizumab outcome, we observed that the proportion of patients with CB is notably higher in the subgroup of TMB-high and PGA-low (5 out of 6) compared to the subgroup of TMB-low and PGA-high (2 out of 15) patients ($p = 0.006$, Fisher's exact test) (see detailed analysis in Supplementary Information and Supplementary Fig. 6).

To identify immune evasion features associated with therapeutic resistance to pembrolizumab, we compared the frequencies of LOH events in HLA-I presentation pathway genes in baseline tumors from HS/CB and LS patients. The frequency of *B2M* LOH was higher in LS compared to HS/CB tumors (12/24, 50% vs 2/16, 13%; Fig. 2A). *B2M* LOH was not detected in HS tumors (0%, 0/9; Fig. 2D). LOH events in other HLA-I presentation and interferon-gamma pathway genes were found infrequently and in similar proportions in both groups. While we have reported a reduced CB rate in patients with germline heterozygous HLA-C locus[8], we did not observe notable associations between the frequency of somatic LOH events in HLA class I genes and pembrolizumab therapeutic benefit. Similar to the results observed in longitudinal biopsies in ICB treated melanoma[15], we observed no significant association between somatic SNV load within these pathways and CB or molecular sensitivity to pembrolizumab.

Lastly, we examined the frequency of somatic copy number alterations (SCNA) associated with regulation of anti-tumor immune responses in pre-clinical and clinical studies: gains in *MYC*[16], losses of *PTEN*[17] and gains or losses in interferon pathway genes (*PDCD1, STAT1, JAK1,* and *JAK2*) (Fig. 2A). *MYC* was gained in 17% (5/29) of LS and only 5% (1/19) of HS/CB. Homozygous loss of *PTEN* was only detected in one LS patient, while single copy loss of *PTEN* occurred in 10% (3/29) of LS and 21% (4/19) of HS/CB patients. Genomic loss of *STAT1, JAK1, JAK2,* or *CD274* were detected infrequently in our cohort. However, it is notable that loss of *STAT1, JAK2,* and *CD274* was detected in a patient with HPV18-positive anal squamous cell carcinoma who did not benefit from treatment.

**Pre-existing immunological activity and immune cell composition as predictors of pembrolizumab clinical response and sensitivity in solid tumors**. To examine whether pre-existing transcriptional signatures of immunological activity and immune cell subpopulation composition in the tumor are associated with pembrolizumab sensitivity and clinical response, we investigated previously published gene-expression signatures for total immune infiltration (IM; inferred total immune infiltration score)[18], interferon-gamma signaling (IFNG)[19], cytolytic activity (CYT)[5], and abundances of 22 immune cell populations inferred by CIBERSORT deconvolution analysis in 65 baseline tumors (CB rate = 30%).

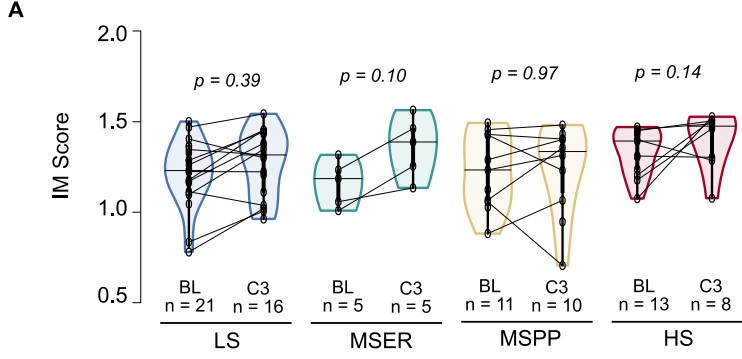

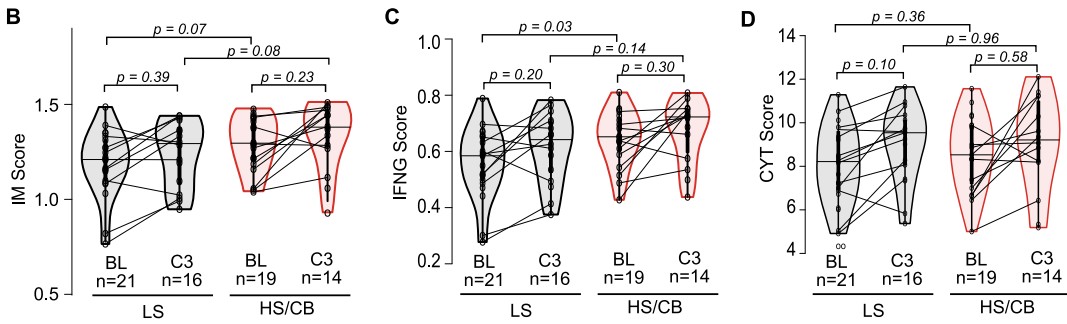

**E**  uncorrected p < 0.10

| Signature | HS/CB mean (range) | LS mean (range) | | Delta-mean (95% CI) | p |
|---|---|---|---|---|---|
| IMS | 1.3 (1.05-1.52) | 1.21 (0.77-1.49) | | 0.09 (-0.01,0.2) | 0.066 |
| CYT | 8.68 (5.01-11.62) | 7.96 (4.39-11.28) | | 0.67 (-0.35,1.94) | 0.178 |
| IFNG | 0.65 (0.43-0.82) | 0.57 (0.19-0.79) | | 0.08 (0,0.15) | 0.036 |
| B.cells.naive | 1.77 (0-8.67) | 0.97 (0-10.3) | | 0.1 (-0.08,0.43) | 0.313 |
| B.cells.memory | 0.41 (0-2.12) | 0.16 (0-2.21) | | 0 (0,0.01) | 0.352 |
| Plasma.cells | 0.33 (0-2.29) | 0.56 (0-3.53) | | -0.13 (-0.48,0.03) | 0.086 |
| T.cells.CD8 | 4.33 (0-13.99) | 2.39 (0-9.9) | | 0.74 (-0.49,3.65) | 0.247 |
| T.cells.CD4.naive | 0.08 (0-1.39) | 0.03 (0-0.73) | | 0 (0,0) | 0.7 |
| T.cells.CD4.memory.resting | 2.84 (0-13.14) | 2.68 (0-12.33) | | -0.11 (-1.17,1.13) | 0.874 |
| T.cells.CD4.memory.activated | 1.18 (0-8.51) | 0.65 (0-4.72) | | 0 (0,0.48) | 0.308 |
| Tfh | 1.75 (0.07-4.18) | 1.28 (0.1-7.35) | | 0.51 (-0.22,1.35) | 0.203 |
| Tregs | 0.06 (0-0.37) | 0.04 (0-0.5) | | 0 (0,0) | 0.59 |
| Tgd | 0.14 (0-1.16) | 0.2 (0-4.73) | | 0 (0,0) | 0.528 |
| NK.cells.resting | 0.09 (0-1.29) | 0.1 (0-1.44) | | 0 (0,0) | 0.578 |
| NK.cells.activated | 1.81 (0-5.85) | 0.94 (0-3.43) | | 0.4 (-0.16,1.33) | 0.201 |
| Monocytes | 0.59 (0-2.49) | 0.66 (0-3) | | 0.01 (-0.27,0.37) | 0.913 |
| Macrophages.M0 | 1.11 (0-5.33) | 1.1 (0-7.68) | | 0 (-0.35,0.74) | 0.881 |
| Macrophages.M1 | 1.84 (0.05-5.91) | 0.89 (0-5.1) | | 0.44 (-0.17,1.38) | 0.141 |
| Macrophages.M2 | 3.93 (0.47-11.89) | 3.47 (0.67-8.63) | | 0.29 (-1.16,1.55) | 0.782 |
| Dendritic.cells.resting | 0.45 (0-2.29) | 0.28 (0-3.49) | | 0 (-0.06,0.07) | 0.688 |
| Dendritic.cells.activated | 0.23 (0-2.41) | 0.33 (0-4.86) | | 0 (-0.12,0) | 0.741 |
| Mast.cells.resting | 0.39 (0-6.81) | 0.16 (0-3.86) | | 0 (0,0) | 0.133 |
| Mast.cells.activated | 3.86 (0-6.88) | 4.83 (0-16.64) | | -0.03 (-1.61,1.42) | 0.99 |
| Eosinophils | 0.2 (0-1.56) | 0.23 (0-1.2) | | 0 (-0.11,0.07) | 0.797 |
| Neutrophils | 0.08 (0-0.43) | 0.24 (0-1.43) | | 0 (-0.06,0.05) | 0.929 |

-1.5 0 1 2 3
$Mean_{HS} - Mean_{LS}$

The signatures of immunological activity varied by cancer types and their distributions were largely overlapping (Supplementary Fig. 7A). Significant correlation was observed between all three signatures, with the strongest concordance between IM and IFNG scores (Spearman correlation = 0.92, $p < 0.01$) (Supplementary Fig. 7C). When comparing signatures across sensitivity strata (Fig. 3A and Supplementary Fig. 7B), we observed the medians of IM, IFNG, and CYT scores in the HS group were highest amongst all sensitivity groups. IFNG ($p = 0.03$, two-sided Wilcoxon rank-sum test; Fig. 3C) and IM ($p = 0.07$, two-sided Wilcoxon rank-sum test; Fig. 3B, E) scores were higher in HS/CB group compared to LS, while no difference was observed in the CYT score (Fig. 3D, E). Patients with IFNG scores greater than the cohort median (IFNG = 0.61) had improved OS (HR = 0.30,

**Fig. 3 Pan-cancer assessment of gene-expressed based immune scores as predictive biomarkers for pembrolizumab. A** Distribution of immune (IM) score within each pembrolizumab sensitivity group. Scores derived from baseline and cycle 3 tumor samples are shown. Lines connect samples collected from the same patient. *P*-values were calculated by two-sided Wilcoxon rank-sum tests. **B** Comparison of IM, **C** Interferon gamma (IFNG), and **D** cytolytic (CYT) scores between LS and HS/CB groups at baseline and cycle 3 of treatment. For all violin plots in **A–D**, the median for each group is shown as a solid horizontal line dot and data points are shown as black dots. The distance between the third-quartile (Q3) and first-quartile (Q1), known as the interquartile range (IQR), is marked around the median by a black rectangle. Vertical lines extending from the top and bottom of the rectangle show the maximum (Q3 + 1.5-times IQR) and minimum (Q1 + 1.5-times IQR). Statistical significance was determined using two-sided Wilcoxon rank-sum tests between indicated sample groups. **E** Forest plot summaries of comparisons between HS/CB and LS baseline tumor gene-expression derived immune activity/ infiltrating immune cell signatures. Two-sided Wilcoxon rank-sum tests were performed to assess statistical significance of the observed differences in signatures between groups. For each score, the difference between group means (Mean$_{HS/CB}$−Mean$_{LS}$) is shown as a solid dot with whiskers indicating the 95% confidence interval. *P* values are uncorrected for multiple testing. Source data are provided in SourceData_Fig3.xlsx. LS low sensitivity, HS/CB high-sensitivity/clinical benefit.

log-rank $p < 0.001$) and PFS (HR = 0.48, log-rank $p < 0.01$) (Supplementary Fig.8).

We observed good correlations between flow cytometry and estimated abundances in B-cells ($R = 0.74$, $p < 0.001$) and CD8 + T-cells ($R = 0.61$, $p < 0.001$) in 51 tumor samples with matching flow-cytometry data[7] (Supplementary Fig. 9). While we detected elevated levels of tumor-infiltrating immune cells with tumor recognition potential such as CD8 T-cells, T follicular helper cells (Tfh), activated natural killer (NK) cells, and M1 macrophages in HS/CB patients when compared LS patients, the findings were not statistically significant ($p > 0.10$, two-sided Wilcoxon rank-sum test, Fig. 3E). Plasma cell score was notably lower ($p = 0.09$, two-sided Wilcoxon rank-sum test, Fig. 3E) in HS/CB patients compared to LS patients. When we evaluated the association between cell scores and CB, PFS, and OS, we found that high Tfh was significantly associated with favorable outcomes (log-rank $p < 0.10$, unadjusted for multiple testing, Supplementary Fig. 8). Conversely, high neutrophil levels were associated with resistance and diminished overall survival (HR = 0.30, log-rank $p = 0.09$, unadjusted for multiple testing, Supplementary Fig. 8). We used the top tertile (>66-percentile) scores of each cell type to define the high signature comparison sample group. Elevated levels of cell types previously observed in the pembrolizumab sensitivity comparison (CD8 + T cells, CD4 + T-cell populations including T regulatory cells (Tregs), activated NK cells and M1 macrophages) were notably associated with overall survival (log-rank $p < 0.10$, unadjusted for multiple testing, Supplementary Fig. 8). Here, we conclude that measurements of spontaneous immunological activity and infiltrate composition differ between individual tumors and may have potential to predict patient outcome to anti-PD1 treatment.

**Pembrolizumab therapy induces immune microenvironment sculpting in solid tumors.** We hypothesized that modulations in the immunological activity as a pharmacological consequence of pembrolizumab therapy may be reflected by changes in the transcriptional signatures and immune cell tumor infiltration associated with immune responses. In the subset of 43 patients with available paired baseline and on-treatment tumor gene-expression profiles, we observed increased IM, IFNG, and CYT scores after pembrolizumab treatment in 70% of the patients (Fig. 3A and Supplementary Fig. 10). Notably higher levels of CD8 + T-cells, CD4 + memory resting T-cells, Tfh, gamma-delta T-cells, M1 macrophages, and eosinophils were observed in tumors post-pembrolizumab therapy compared to baseline ($p < 0.10$, unadjusted for multiple testing, two-sided Wilcoxon rank-sum test) (Supplementary Fig. 10). Our results are consistent with increased immune cell populations in melanoma tumors in response to nivolumab treatment[20]. Here, we demonstrate that regardless of tumor type, anti-PD1 therapy modulates both the immune response and immune cell repertoire within the tumor tissue.

**Gene expression profiling in longitudinal tumor biopsies identifies immune regulatory factors in patients treated with pembrolizumab.** To identify the differences in gene expression changes between matched baseline and on-treatment tumor samples that could differentiate HS/CB ($n = 11$) from LS ($n = 11$) patients, we performed a supervised differential gene expression analysis using a negative binomial generalized model design that included control for patient-specific variations. Within the HS/CB group, we detected a robust change in tumor transcriptional profiles characterized by 1443 differentially regulated genes (DRG) (FDR-adjusted $p$-value ≤ 0.10) with large effect sizes (log2 fold-change from baseline range = −28–29) (Fig. 4B and Supplementary Data 3). In contrast, few changes in tumor gene-expression (185 genes differentially regulated, log2 fold-change from baseline range = −1.8–2.5) were observed in LS tumors (Fig. 4B and Supplementary Data 4).

We focused on the 57 DRGs shared by both groups and observed that 93% (53/57) were up-regulated following treatment (Fig. 4A and Supplementary Data 5). A STRING[21] protein-protein interaction (PPI) network analysis of the up-regulated DRGs identifies positive regulation of immune process and T-cell activation amongst the enriched molecular functions.

We hypothesized that genes with larger expression increase post-ICB in LS characterize the activity of immune evasion factors while those in HS/CB reflect anti-tumor activity or emerging adaptive resistance to anti-PD1. Within the DRGs more highly induced in LS than HS/CB, we found that 7 of 8 DRGs (*CHI3L1, CLEC4E, CXCL9, VNN2, GBP2, TLR8,* and *ADAM-DEC1*) were abundantly expressed by immune-suppressive myeloid cells[22–28]. This suggests that recruitment or activation of immune-suppressive myeloid cells contribute to pembrolizumab resistance in solid tumors. Within the 45 DRGs with higher level of induction in HS/CB, we identified CD8 marker genes (*CD8A* and *CD8B*), IFN gamma response genes (*PYHIN1, CXCR6, CXCL13, CXCL11, WAS, GBP5*) inhibitory checkpoint molecules (*PDCD1, TIGIT, CTLA4*)[1] (Fig. 5A, C), and cytolytic molecules with anti-tumor activity (*GZMA, GZMB*)[5]. Interestingly, in both LS and HS/CB samples we identified increased expression of *PLA2G2D* (Fig. 4D), with a trend towards higher magnitude of increase in the HS/CB group. *PLA2G2D* is a secreted phospholipase with experimental evidence to suggest its function to attenuate T helper 1 immune responses by maintaining the steady-state levels of anti-inflammatory lipids in murine lung tissues during viral infections[29]. In a pan-cancer analysis with TCGA data of primary tumors[5], *PLA2G2D* expression was correlated with CYT, to a similar degree with other immunosuppressive factors and interferon-stimulated T-cell attracting cytokines. We further validated its up-regulation in an independent dataset of 62 metastatic melanoma tumors upon nivolumab treatment[20] ($p = 0.078$ two-sided Wilcoxon rank-sum test). We also found evidence that responders to anti-PD1 or anti-

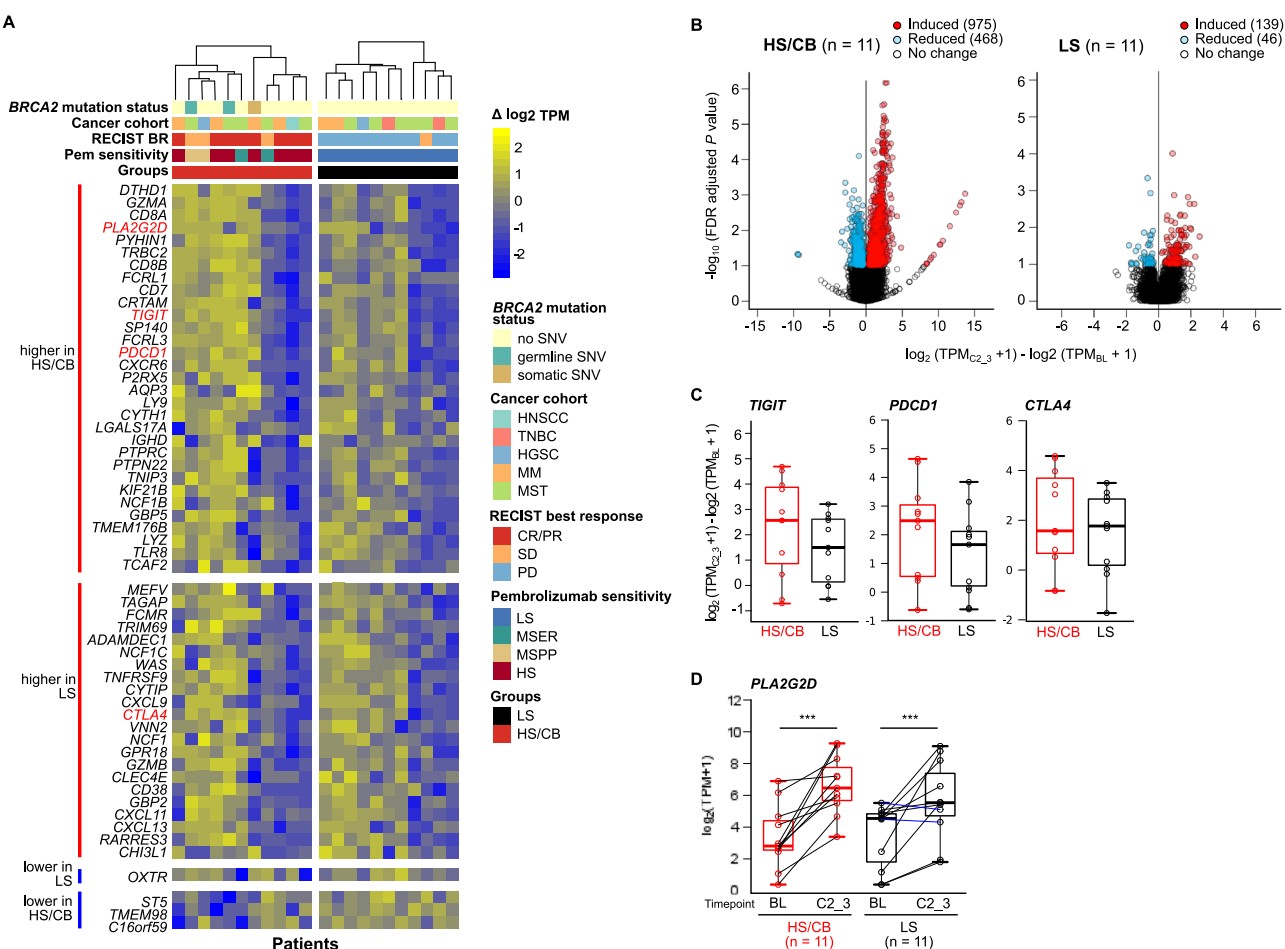

**Fig. 4 Genes differentially regulated by pembrolizumab treatment in treatment sensitive and resistant tumors. A** Heatmap of differentially regulated genes by pembrolizumab treatment to differentiate treatment sensitive from insensitive tumors. Samples are grouped by HS/CB or LS and sorted by hierarchical clustering within each group based on the similarity of change in gene-expression of selected genes. Genes are organized into four groups according to the combination of gene-regulation and differences between the two comparison groups. From top to bottom: genes up-regulated in both groups with higher expression in the HS/CB group, genes up-regulated in both groups with higher expression in the LS group, genes down-regulated in both groups with a greater degree of down-regulation in LS, and genes down-regulated in both groups with a greater degree of down-regulation in HS/CB. Genes in each group are listed in decreasing order of the difference of median change in gene-expression between HS/CB and LS groups. **B** Volcano plots of differentially expressed genes comparing paired early on-treatment to baseline tumor gene-expression in HS/CB or LS groups. Statistical significance for gene selection: FDR-adjusted *p*-value ≤ 0.10. **C** Distributions of change in gene-expression of immune checkpoint genes (*TIGIT*, *PDCD1*, *CTLA4*) in HS/CB (*n* = 11) versus LS (*n* = 11) baseline and on-treatment tumor pairs. The median for each group is marked by a horizontal line in the center of a rectangle and data points are shown as open circles. The distance between the third-quartile (Q3) and first-quartile (Q1), known as the interquartile range (IQR), is marked around the median by a rectangle. Vertical lines extending from the top and bottom of the rectangle show the maximum (Q3 + 1.5-times IQR) and minimum (Q1 + 1.5-times IQR). **D** *PLA2G2D* gene expression before and after pembrolizumab treatment in HS/CB (*n* = 11, *p* = 0.001) or LS (*n* = 11, *p* = 0.005) tumors before and after 2–3 cycles of pembrolizumab treatment. *P*-values were calculated by two-sided Wilcoxon rank-sum tests and uncorrected for multiple testing. The median for each group is marked by a horizontal line in the center of a rectangle and data points are shown as open circles. The distance between the third-quartile (Q3) and first-quartile (Q1), known as the interquartile range (IQR), is marked around the median by a rectangle. Vertical lines extending from the top and bottom of the rectangle show the maximum (Q3 + 1.5-times IQR) and minimum(Q1 + 1.5-times IQR). Source data are provided in SourceData_Fig4.zip. LS low sensitivity, HS/CB high-sensitivity/clinical benefit, FDR false discovery rate.

PDL1 had higher *PLA2G2D* expression in pre-therapy tumors compared to non-responders in urothelial cancer[30] and anti-CTLA4 pre-treated advanced melanoma[20] data sets (*p* < 0.10, uncorrected for multiple testing, Supplementary Fig. 11)[31]. Together, this data suggest the potential of *PLA2G2D* as a biomarker of tumor immunity.

**Differential regulation of T- and B-cell activation and signaling transcription programs in anti-PD-1 treated tumors is predictive of clinical response.** To identify the molecular processes that are differentially regulated by pembrolizumab

treatment between HS/CB and LS, we performed gene-set enrichment analysis[32] of Gene Ontology (GO) biological processes to derive biological meaning of the genes differentially regulated by pembrolizumab between HS/CB and LS. We identified 93 significantly enriched GO terms (FDR-adjusted *p* < 0.05) in genes with increased abundance following treatment in HS/CB compared to LS, and 104 terms enriched in genes with increased abundance in LS compared to HS/CB (Fig. 5A and Supplementary Data 6). Processes such as B-cell receptor signaling pathway, immunoglobulin production, T-cell activation, and migration were up-regulated in HS/CB versus LS, reflecting a high level of pembrolizumab treatment-induced immune cell activity in the

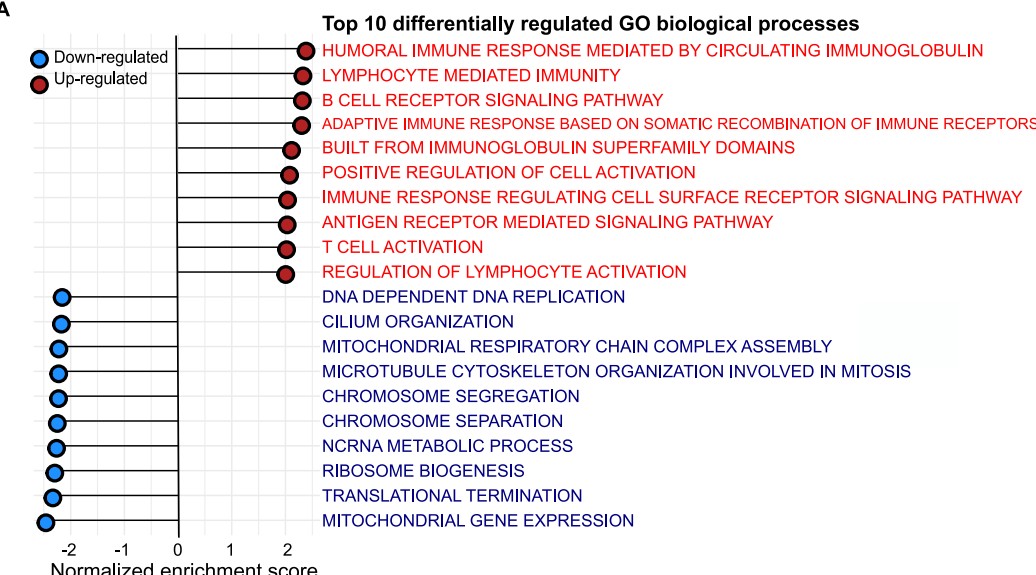

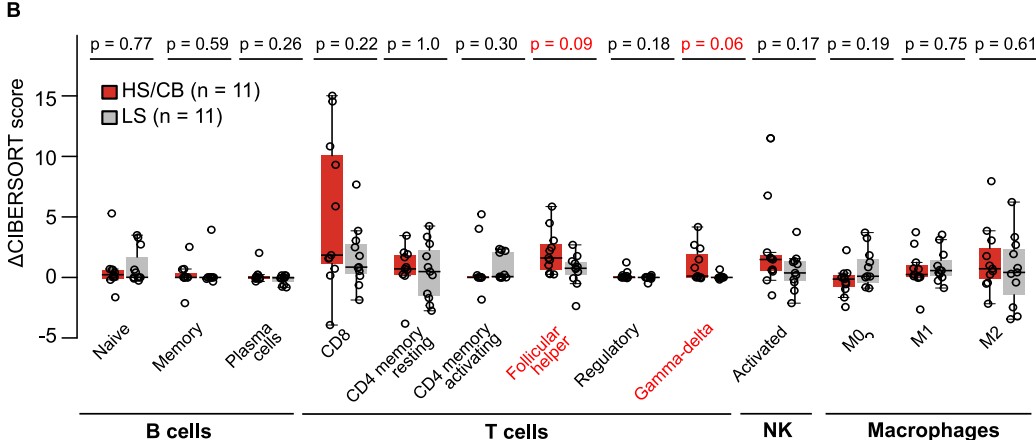

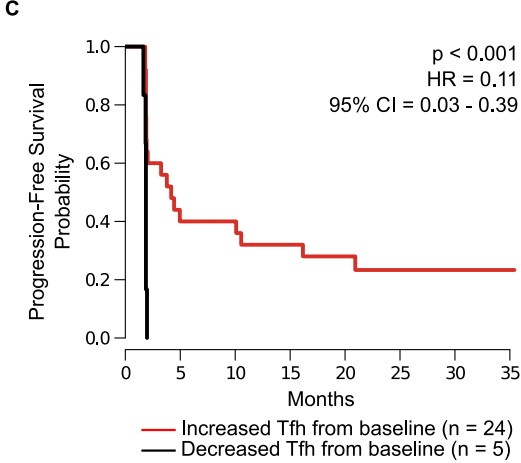

TME. Unsurprisingly, processes associated with DNA replication and mitosis were up-regulated in LS, possibly reflecting continued growth of malignant cells resistant to pembrolizumab treatment.

Next, we asked whether the observed changes in gene expression profiles are associated with changes in abundance of the specific cell types within the TME. Guided by the GO enrichment analysis results, we compared the change in cell scores obtained from CIBERSORT deconvolution for B-cell (n = 3), T-cell (n = 6), NK cell (n = 1), and macrophage (n = 3) subsets between HS/CB and LS. While we observe the largest increases in multiple T-cell subsets in HS/CB versus LS (Tfh and gamma-delta T-cells, two-sided Wilcoxon rank-sum test uncorrected p value < 0.10), we did not observe increases in B-cell score (Fig. 5B). We also observed a trend towards increased CD8 + T

**Fig. 5 Changes in immune cell composition in the TME associated with pembrolizumab sensitivity. A** Top 10 enriched down-regulated and up-regulated Gene Ontology Biological Processes from genes differentially regulated by pembrolizumab in treatment-sensitive compared to treatment-insensitive tumors. **B** Comparison of change in immune cell abundance in the TME at cycle 3 of pembrolizumab treatment between sensitive and insensitive tumors. The median for each group is marked by a horizontal black line in the center of a rectangle and data points are shown as open circles. The distance between the third-quartile (Q3) and first-quartile (Q1), known as the interquartile range (IQR), is marked around the median by a rectangle. Vertical lines extending from the top and bottom of the rectangle show the maximum (Q3 + 1.5-times IQR) and minimum (Q1 + 1.5-times IQR). *P*-values were calculated by two-sided Wilcoxon rank-sum tests. **C**. Kaplan-Meier plot of progression-free survival in patients stratified by increase or decrease in Tfh score from baseline. (*p* = 0.0007). *P*-values reflect the outcomes of a two-sided log-rank test of a univariate Cox proportional hazards model. Source data are provided in SourceData_Fig5.xlsx. TME tumor microenvironment.

cell scores in HS/CB versus LS, however, the range of the score was large and overlapped substantially between groups. A survival analysis revealed that increased Tfh score early on-treatment was associated with improved PFS (HR increase from baseline = 0.11, *p* = 0.0007, Cox proportional hazards log-rank test) (Fig. 5C). This finding indicates that early change in Tfh score may have predictive value in the pembrolizumab treatment setting.

## Discussion

Biomarkers correlated with pembrolizumab clinical response often lack robust concordance between studies due to poorly defined response subgroups[9,20,33–35]. Here, we show that changes in ctDNA and tumor measurement by imaging detected from baseline as early as 6 to 7 weeks from the start of treatment can stratify patients into subgroups with distinct survival outcomes. We leveraged this approach to strengthen the sample selection criteria of response for molecular biomarker assessments. We demonstrated that patients with high molecular sensitivity to pembrolizumab had the highest TMB, baseline total immune score, and interferon-gamma signaling pathway gene expression levels. This association was observed across tumor types without the need to pre-define tumor-type specific cutoffs. Tumors with low sensitivity were differentiated from patients with mixed response, reducing molecular heterogeneity of the non-responder group. The application of ctDNA dynamics complements existing studies that suggest that assessment of tumor responses using tumor or blood specimens collected early during the course of ICB treatment is highly predictive of overall clinical response[20,35,36] and highlights the importance of longitudinal sample collection for translational research.

Using baseline tumor mutation and gene-expression profiles, we identified mutated and copy number altered genes, and gene-sets associated with clinical benefit and high molecular sensitivity to pembrolizumab that are in agreement with a recent analysis of anti-PD1 treated cancers[9,20]. In cancer studies, *BRCA2* is often investigated in conjunction with *BRCA1* for their DNA repair tumor suppressor function and association with breast and ovarian cancer risks[37]. Our results demonstrate that non-synonymous mutations in *BRCA2*, but not *BRCA1*, were more frequently observed in patients with clinical benefit and early molecular sensitivity. In a recent retrospective analysis of the pan-cancer MSKCC-IMPACT patient cohort, Samstein et al.[38] identified clinical benefit in 44% of patients with BRCA2-deficient cancers following ICB treatment. Our finding, not only supports previous reports of improved response to single-agent ICB in *BRCA2*-deficient tumors, but also emphasizes the need to evaluate *BRCA1* and *BRCA2* mutated patients separately in future clinical trial designs. Due to the limited sample size, subset analysis of germline or somatic SNVs was not explored. Further studies are warranted to examine the functional effect of specific germline *BRCA2* mutations on the activity of immune cells as well as the impact of *BRCA2* mutations on the immunogenicity of tumor cells.

While we observed several mutations in genes related to antigen presentation and interferon-gamma pathways, their presence or frequencies did not provide predictive information for ICB treatment outcomes in our cohort. This is consistent with a study of these features restricted to metastatic melanoma[15]. Our observation of increased frequency of *B2M* LOH events in baseline tumors of LS patients provides evidence to support B2M as an essential protein component for HLA Class I antigen presentation that is highly susceptible to immunoediting during tumorigenesis. This association between *B2M* LOH events and ICB resistance has been demonstrated in three independent melanoma cohorts[9,15,33]. The authors postulate that *B2M* LOH may serve as the initial step towards complete loss of B2M triggered by a second mutation or dysregulation event and provided experimental evidence to demonstrate that cancer cells lacking *B2M* expression are susceptible to elimination by NK cells[15]. Our data illustrate the importance of this mechanism outside melanoma and the need to evaluate B2M LOH as exclusion criterion for ICB or prioritization for NK cell therapy.

To better understand how tumors adapt to immune modulation by anti-PD-1 in treatment sensitive and resistant individuals, we profiled and compared changes in gene expression and TME immune cell composition in tumor samples collected before and after treatment. As expected, we observed up-regulation of known inhibitory immune checkpoints and markers of progressive T cell exhaustion[39] (*PDCD1, TIGIT, CTLA-4*), all of which have demonstrated anti-tumor activity when inhibited alone or in combination with PD-1 inhibitors in various cancer indications[15,35,40]. We also identified increased levels of genes related to CD8 + cytotoxic T lymphocyte activity and interferon-gamma signaling in patients with improved clinical outcomes.

We identified and validated increased phospholipase *PLA2G2D* expression in tumors upon anti-PD-1 treatment. While the bulk of current literature has characterized the role of Pla2g2d in fatty-acid metabolism within the contexts of autoimmune disorders and viral infections[41–43], little is known of its relevance in cancer development and immunotherapy response. Rooney et al.[5] reported that *PLA2G2D* expression is correlated with the cytolytic activity score to suggest its participation in counter-regulatory activities that limit immune responses in primary solid tumors[5]. Moreover, Miki et al.[44], in a melanoma mouse model, demonstrated that Pla2g2d deficiency is associated with delayed tumor growth. Our findings along with growing literature, together suggest that future investigations to elucidate the mechanistic role of *PLA2G2D* in tumor contexts will enrich our understanding of tumor immunobiology.

Finally, we observed increased B- and T-cell activation associated with favorable response and survival benefit in patients with increased expression of the Tfh gene signature after pembrolizumab treatment. It has been demonstrated in murine models of breast cancer that B-cell activation by ICB depended on the activity of Tfh within the germinal centers of spatially organized tertiary lymphoid structures (TLS)[45]. This finding was further corroborated by recent studies reporting prognostic and

predictive value of B-cell signatures and TLS abundance in MM, NSCLC and soft tissue sarcoma treated with ICB[39,46–48]. Collectively, these findings lead us to consider an in-depth evaluation of predictive and prognostic value of TLS abundance and therapeutic strategies to promote TLS formation and B-cell activity in the design of future checkpoint blockade clinical trials.

Together, this study illustrates the need for integrating comprehensive genome and transcriptome profiling of tumor specimens to capture the diversity of cancer genome and immunogenomic mechanisms to complement molecular-based sensitivity measurements using circulating tumor DNA to monitor response to contemporary immunotherapies. An increasing number of clinical trials are being performed evaluating ICB in combinations with chemotherapy, molecularly targeted therapy, and other immunotherapeutic agents[49]. As such, delineation of the molecular and immune milieu before and after ICB monotherapy is crucial to provide a benchmark against which the effects of additional anti-tumor agents can be investigated.

## Methods

**Clinical trial and subject details**. One-hundred and six patients were accrued from 21 March 2016 to 9 May 2018 to a single-center, investigator-initiated phase II interventional clinical trial (NCT02644369, registered here: https://clinicaltrials.gov/ct2/show/NCT02644369) to interrogate the pharmacodynamic activity of pembrolizumab in metastatic solid tumors. The clinical trial was approved by the Research Ethics Board at University Health Network in Toronto, Canada. The trial was conducted in accordance with the principles of Good Clinical Practice, the provisions of the Declaration of Helsinki, and other applicable local regulations. All patients gave their written informed consent. Patients were prospectively enrolled into one of five cohorts: metastatic head and neck carcinoma, triple negative breast cancer, high-grade-serous ovarian cancer, melanoma, and other rare solid tumor types. Key inclusion and exclusion criteria have been summarized and published previously[8]. The study protocol is provided in Supplementary Note 1. All patients received pembrolizumab (200 mg/kg every 3 weeks intravenously) until disease progression or for a maximum of 2 years supplied in kind by Merck. More detailed description of the clinical selection criteria has been described previously[7]. Response assessment by radiographic imaging was performed every 9 weeks (3 cycles of treatment) until progression is confirmed. RECIST v1.1 criteria[50] was used to determine tumor responses for patients at each measured radiographic time point. Clinical benefit was defined by best overall response with complete response, partial response, or stable disease >6 cycles (18 weeks). All patients underwent biopsy before starting therapy and underwent a repeat biopsy, collected from the same site, on cycle 2 or 3 of treatment. Tumor tissue was divided into two for formalin-fixed paraffin-embedded treatment and immediately digested into single cells (Mitenyl GentleMACs System). Tumor cells were flash frozen and stored at −80 °C for DNA/RNA co-extraction (Qiagen DNA/RNA co-isolation kit). A detailed breakdown of tumor types and RECIST response and clinical benefit is provided in Supplementary Data 1.

**Longitudinal ctDNA Assessment**. WES-generated patient-specific tumor somatic mutation profiles were used to design bespoke ctDNA assays by Natera Inc. (San Carlos, USA) using their proprietary Signatera™ assay. The assay uses calculated mutation frequencies and estimated clonality to select and design multiplex PCR primers targeting 16 highly-ranked tumor variants in each patient[51–54]. Illumina sequencing platform was used to perform amplicon deep sequencing of products obtained from targeted PCR. For each baseline and on-treatment time-point plasma sample, ctDNA was quantified in units of mean tumor molecules (MTM) per mL of plasma. This takes into account mean allele frequencies across all mutations, cfDNA extracted, and plasma volume. The early change in ctDNA was calculated as the percentage difference in absolute ctDNA levels between the second or third treatment cycle and baseline time-points. An increase was defined greater than zero, while decrease was <0.

**Whole-exome sequencing**. DNA from tumor and pre-therapy peripheral blood mononuclear cells were extracted from frozen cell pellets stored at −80 °C (Qiagen DNA/RNA co-isolation kit). Agilent SureSelect V5 + UTR probes were used for hybrid selection to enrich for exonic sequences. Prepared libraries were sequenced with paired-end 125 bp reads on the Illumina HiSeq2000 or 2500 platform per manufacturer's protocols at the Princess Margaret Genomic Centre or Translational Genomics Laboratory in Toronto, ON. Tumor samples were sequenced to a median depth of 250x and normals to a median depth of 50x. Sequence alignment to human reference genome version hg38 using the Burrows-Wheeler Alignment tool (v.0.7.12), co-cleaning, and duplicate removal was performed[7,55]. Pathogenic germline mutations were identified using GATK HaplotypeCaller (GATK

v.4.0.5.1)[56] and overlapped with mutations annotated as having reported evidence for likely pathogenic consequence contributing to cancer within the ClinVar database (v022019)[57]. Candidate germline mutations occurring in more than 4 subjects within the entire cohort were not considered to be truly pathogenic. Somatic single-nucleotide variations (SNVs) were identified using a combination of 5 mutation callers (Mutect 2 GATK v.3.8[58], Mutect v1.1.4[59], Strelka v1.0.14[60], Varscan v2.4.2[61], and Vardict v1.5.8[62]). Small insertions and deletions (indels) were identified using a combination of 4 mutation callers (Mutect 2 GATK v.3.8[58], Strelka v1.014[60], Varscan v2.4.2[61], and Vardict 1.5.8[62]). All mutations were annotated with Variant Effect Predictor (v92)[63] and filtered with dbSNP (v150)[64], gnomAD (v170228)[65] to remove variants likely to be germline. Tumor mutation burden was defined as the number of remaining non-synonymous mutations per million bases covered by supporting sequencing reads[20,66]. Contributions of mutational signatures in COSMIC[67] were determined in each sample using non-negative least-squares regression provided by the deconstructSigs v1.8.0 R package[68]. Analysis and visualization of mutations were performed using maftools R package[69].

Data quality control was assessed using Picard metrics (v.2.10.9) and genotype matches between normal/tumor sample pairs using NGSCheckMate [https://parklab.github.io/NGSCheckMate][70].

**Mutation enrichment analysis**. To identify differentially mutated genes in HS/CB or LS tumors, we selected candidate genes that are recurrently mutated in at least 25% of the samples in one group (5 out of 21 in HS/CB or 13 out of 51 in LS) and at most 5% in the other (1 out of 21 in HS/CB or 3 in LS). We considered only non-synonymous somatic events (missense, nonsense, non-stop, translational start site, splice site, in-frame, and frame-shift insertions and deletions) and pathogenic germline mutations in this analysis. Using these criteria, a total of 49 genes were selected (47 enriched in HS/CB and 2 enriched LS) and tested for occurrence higher than the estimated background mutation rate (Fisher exact test, FDR-corrected, $p < 0.05$). To account for the effects of uneven distribution of cancer histologies in the HS/CB and LS groups within our data, the background mutation rate was estimated for each gene of interest as the weighted sum of mutation rates from cancer types with WES sequencing data made available through the TCGA PanCancer Project (30 solid cancer types, $n = 10,195$) (https://cbioportal.org).

**Copy number analysis**. Tumor cellularity, ploidy, and allele-specific DNA copy number for each sample was determined from WES using the Sequenza R package[71] with supplied cancer-type specific ploidy prior calculated from TCGA. Manual inspection was performed to select the most likely model-fit from the top 5 solutions selected by Sequenza. Samples with estimated cellularity of <20% are excluded from further analyses. The percent of copy-number altered genome was defined as the percentage of the genome with non-diploid total copy-number. Loss-of-heterozygosity was defined as genomic regions with a total copy number of 1. Copy number gain is defined as greater than tumor ploidy + 2.

**HLA class I typing, mutation, and LOH detection**. Class I HLA types for each patient were first inferred from the germline WES data and patient ethnicity information using PolySolver[72] to 4-digit resolution. Using the patient-specific HLA type information, somatic mutations, and loss-of-heterozygosity in HLA class I genes were identified in tumor samples using PolySolver[72] and HLA-LOH [https://bitbucket.org/mcgranahanlab/lohhla/src/master/][73].

**Microsatellite instability status prediction**. Microsatellite instability status for each sample was determined from WES using mSINGS [https://bitbucket.org/uwlabmed/msings.git][74] with default threshold and cutoffs. The pooled normal baseline is generated from the distribution of unique alleles at the 2539 loci for analysis of WES TCGA data from all patient normal samples as provided by the mSINGS software.

**RNA sequencing**. Extracted RNA from tumor cells were sequenced using previously published protocols[55] at the Princess Margaret Genomic Centre and Translational Genomics Laboratory. FASTQs were aligned to the hg38 human reference genome using STAR 2.4.2a[75] aligner with default settings. Data quality was assessed using RNA-seQC (v.1.1.8). Expression levels of all transcripts were quantified using RSEM 1.3.0[76] with the GENCODE transcript reference version 26. RNA data quality metrics were collected using RNAseQC v1.1.8[77] on genome aligned and duplicate-marked BAM files. Upon manual inspection of principal component analysis of quantile normalized log2-transformed gene-expression data, outliers corresponding to high transcript 5' to 3' coverage bias, strand-specific bias, and low gene counts, were removed from downstream analysis. Batch effects arising from technical differences between sequencing facilities were normalized using sva R-package (v.3.36)[78] implementation of ComBat[79] on log2-transformed quantile normalized data across all remaining samples. RSEM quantified normalized read counts per transcript were used as input for differential expression analyses. Log2-transformed and batch-normalized gene-expression in transcripts per million (TMP) were used for visualizations in heatmaps and boxplots.

**Gene set enrichment scores**. To derive absolute enrichment scores from published and experimentally validated gene signatures of total immune infiltration (ESTIMATE immune score)[18] and IFNG.GS (MSigDB Hallmarks signatures)[19] for each tumor sample, we used the GSVA R package[80] implementation of ssGSEA to calculate single-sample gene set enrichment (ssGSEA) scores. Batch normalized and log2-transformed gene-expression profiles were used as input to ssGSEA.

**Differential gene expression**. DESeq2[81] was used for differential gene expression analysis. We used a pairwise approach controlling for each patient to compare on-treatment (cycle 2 or 3) to baseline expression levels separately in each group (HS/CB and LS). Transcripts with a low mean read count (<10) were excluded from the analysis[36]. Binomial Wald test was used after correcting for size factor and dispersion by applying default DESeq2 parameters. To explore significant differences between HS/CB and LS responses to anti-PD-1 treatment, we tested for interaction between treatment and ICB sensitivity grouping (HS/CB; $n = 11$ or LS; $n = 11$).

**Identification of ICB-regulated genes and gene set enrichment**. To identify genes commonly regulated by ICB treatment between HS/CB and LS groups, we selected genes that were responsive (up or down-regulated) to treatment at FDR-adjusted $p < 0.10$ separately in each group. We then overlapped the two lists of candidate genes to identify the subset genes common in both groups.

Gene set enrichment analysis to identify treatment responsive pathways that are differentially regulated in HS/CB and LS, we ranked genes in order of differential induction (metric $= -\log10(p\text{-value})/(\text{sign of fold-change})$). Gene-set enrichment analysis (GSEA)[82] was performed on the ranked gene-list and metric with the GSEA implemented in R package fgsea[32] and gage[83] using the GO biological processes[84,85] terms and pathways containing 15–500 genes, with 1000 permutations (custom code provided). Pathways and terms at FDR < 0.10 are selected as statistically significant and enrichment scores are visualized using R ggplot2[86].

**TME cell-type inference**. The relative fractions of 22 cell subsets within the TME lymphocyte population were inferred from batch-normalized gene-expression profiles in linear-space for each tumor sample using CIBERSORT v.1.06 [https://cibersort.stanford.edu/][87]. The program was run with the default LM22 reference matrix on absolute-mode estimated abundance of cell subsets and in the TME. Based on prior knowledge of cell-type similarities, we aggregated estimated values of similar cell types to reduce the cell subset complexity in specific analyses.

The log2 fold change for inferred cell type abundances to reflect the dynamic effects of ICB treatment on the TME is calculated in two ways. In patients with paired baseline and on-treatment tumor measurements, the change is calculated as the difference of the log2-transformed absolute values for each cell type between the pre- and on-treatment sample pair. For samples without a baseline tumor measurement, the fold-change is calculated as the difference of the log2 transformed absolute values between the on-treatment sample and median value from all baseline tumors in the study data cohort ($n = 65$).

**Flow cytometry**. Flow cytometry for quantifying CD3 + T- and CD19 + B- cells in the tumor tissue was performed using optimized antibodies and dilutions provided in Supplementary Table 2. Briefly, tumor single-cell suspensions were stained for immune markers of interest. The 5 laser LSR Fortessa X-20 (BD, Mississauga, Ontario, Canada) was used for data acquisition. Marker quantification and analysis were performed using FlowJo v10 (Treestar, Ashland, Oregon, USA). The gating strategy is shown in Supplementary Fig. 9A.

**PD-L1 immunohistochemistry**. Immunohistochemical (IHC) staining for PD-L1 using the mouse monoclonal anti-PD-L1 antibody (clone 22C3 at 2 µg/mL, Merck, Palo Alto, CA) was performed by Qualtek Molecular Laboratories (Newtown, PA, USA). The PD-L1 IHC assay has been previously validated and being used in the Merck Investigator Studies Program[88]. The level of PD-L1 staining is reported by Qualtek as a modified proportion score (MPS, range 0–100), indicating the percentage of PD-L1-expressing tumor cells and mononuclear inflammatory cells within the tumor nest. Briefly, 4–5 µm formalin-fixed, paraffin-embedded baseline tumor tissue sections are mounted on positively charged slides. To prepare the slides for antigen/epitope retrieval and primary antibody staining, slides are baked (60 °C dry heat for 45 min) and de-waxed and rehydrated using a seers of solvent washes in decreasing concentration (4 × 100% xylene, 100%, 70%, 30% ethanol, and distilled water). Antigen retrieval was performed in two steps: first using a low pH Target Retrieval Solution (Dako Cat. No. S1700) at 90 °C for 20 min, followed by 1:160 Proteinase K treatment for 10 min at room temperature (Dako Cat. No. S3020 diluted in EnVision™ FLEX + wash buffer). Prepared slides are incubated off-platform for 16 ± 1 h in a dark humidified chamber with the primary antibody diluted in Primary Antibody Diluent (Dako Cat. No. S0809). The EnVision™ FLEX + reagent system (EnVision FLEX + HRP-Polymer kit (Dako Cat. No. K8012). EnVision™ FLEX + Mouse Linker (15 min), EnVision™ FLEX + HRP-polymer (25 min), EnVision™ FLEX + DAB Chromogen (10 min), nickel chloride (Ni + Cl2) DAB Enhancer (10 min), with hematoxylin counterstain (1/5 dilution, 1 min)) were used for the subsequent wash, block, and signal amplification and detection automated with a TechMate Instrument (Roche Diagnostics, QML workmate v3.96.) at room temperature. PD-L1 IHC reactivity interpretation was evaluated by a board-certified pathologist at QualTek.

**Statistics and survival analysis**. All statistical analyses were performed in the R Statistical Computing Environment v3.3.1 (R Foundation for Statistical Computing, Vienna, Austria. URL https://www.R-project.org/). Custom code to reproduce key figures and results reported in the manuscript are available at https://github.com/pughlab/inspire-genomics. Wilcoxon rank-sum test (for two groups) or Kruskal–Wallis test (for more than two groups) was performed to examine group differences for continuous measures. Wilcox rank-sum tests were performed on continuous measures between two groups to examine differences in distributions. Cox proportional hazards regression models (univariate) were used to assess the impact of pembrolizumab sensitivity groups, TMB, immune gene set scores at baseline and change in immune cell inference scores on OS and PFS. Fisher's exact test was used to assess the enrichment of mutations in a given gene as compared to the background mutation rate. All tests were two-sided with $p \leq 0.10$ considered to be statistically significant. Nominal $p$-values were reported throughout. Multiple testing adjustments using the Benjamini–Hochberg False Discovery Rate method was applied to differential gene-expression and GSEA GO enrichment analyses.

**Reporting summary**. Further information on research design is available in the Nature Research Reporting Summary linked to this article.

## Data availability

Anonymized patient normal, tumor exome, and RNA-seq bam files containing alignments of the original raw sequencing reads used in this study have been deposited in the European Genome-phenome Archive repository under accession code EGAS00001003280. The processed variant calls are available at EGAD00001006569. The datasets are available under restricted access in compliance with patient consent for data sharing, access can be obtained by approval from the University Health Network data access committee (Contact person: Natalie Stickle, Email: natalie.stickle@uhn.ca). A redacted version of the clinical trial study protocol is provided in Supplementary Note 1 in the Supplementary Information file. The publicly available datasets (Broad MSS mixed solid tumors[10], UMich MET500[11], and MSKCC-IMPACT IO study[12]) used in this study are available via cBioPortal [https://www.cbioportal.org/][89,90]. The remaining data, including de-identified clinical data, are available within the Article, Supplementary Information or Source Data file. Source data are provided with this paper.

## Code availability

Custom code for analysis and producing visualization of the paper can be accessed via the project github repository [https://github.com/pughlab/inspire-genomics][91].

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

## Acknowledgements

We would like to thank the patients and their families for their participation and contributions to the INSPIRE clinical trial. Major funding support for the project was made possible by the Princess Margaret Cancer Foundation, Ontario Institute for Cancer Research, and Terry Fox Research Institute. We thank Merck Canada Inc., Kirkland, QC, Canada for contributing the study drug for the clinical trial. S.Y.C.Y. was supported by the Medical Biophysics OSOTF Excellence Award, Ontario Graduate Scholarship, and the Queen Elizabeth II/Heart and Stroke Foundation of Ontario Graduate Scholarships in Science and Technology at the University of Toronto. M.A.J.I. was supported in part by a fellowship through the BMO Chair in Precision Cancer Genomics. L.L.S. holds the BMO Chair in Precision Cancer Genomics. T.J.P. holds the Canada Research Chair in Translational Genomics and is supported by a Senior Investigator Award from the Ontario Institute for Cancer Research and the Gattuso-Slaight Personalized Cancer Medicine Fund. We gratefully acknowledge the individuals from the Princess Margaret Tumor Immunotherapy Program, including the Administrative (Kendra Ross, Helen Chow, Sawako Elston, and Aileen Trang), Correlatives (Sevan Hakgor, Amanda Giesler, Koosha Vakilli), and Immune Monitoring teams (Diana Gray, Valentin Sotov, Diane Liu, Mark Camacho, Darya Lemiashkova, Ramy Gadalla, Reema Deol, Drew Wallace, Douglas Millar, and Babak Noamani) for coordinating receiving, processing, and biobanking tumor and blood samples and performing flow-cytometry experiments. We thank the staff of the Princess Margaret Genomics Centre (www.pmgenomics.ca, Troy Ketela, Neil Winegarden, Julissa Tsao, and Nick Khuu), Bioinformatics Services (Carl Virtanen, Zhibin Lu, and Natalie Stickle), and the Translational Genomics Laboratory ( https://labs.oicr.on.ca/translational-genomics-laboratory, Dax Torti, Kayla Marsh, Jenna Eagles, Alberto Leon, Lawrence Heisler, Jonathon Torchia, Prisni Rath, and Alexander Fortuna) for their expertise in generating the sequencing data used in this study. The Translational Genomics Laboratory is a joint initiative between the Princess Margaret Cancer Centre and the Ontario Institute for Cancer Research that is enabled through funding provided by the Government of Ontario and the Princess Margaret Cancer Foundation. Additional infrastructure support from the Canada Foundation for Innovation, Leaders Opportunity Fund [CFI #32383]; Ontario Ministry of Research and Innovation, Ontario Research Fund Small Infrastructure Program; and Ontario Institute for Cancer Research (genomics.oicr.on.ca). Lastly, we thank members of the Pugh Lab (Arash Nabbi, Danielle Croucher, Laura Richards, Petr Smirnov, Rene Quevedo, and Stephenie Prokopec) for their extensive peer editing and feedback to improve the manuscript.

## Author contributions

P.S.O, L.L.S and T.J.P conceived the study. P.S.O, L.L.S and T.J.P. secured funding. S.Y.C.Y. performed data analysis, statistical analysis, and data visualization. S.C.L., B.X.W. and D.L.C. performed FACs experiments and data analysis. M.A.J.I., M.O. and L.L.S. reviewed and curated the clinical data. A.R.H., P.L.B., S.L., A.S., A.A.R. and L.L.S. accrued patients and supervised biospecimen sample collection with V.S. and H.K.B.; T.J.P., P.S.O., D.L.C. and D.T. supervised biospecimen sample processing by S.Y.C.Y., S.C.L., B.X.W., D.L.C., bY.H., I.C. and S.E.G.; K.Z. and J.P.B. provided technical expertize for bioinformatic analyses and software development. A.A., B.H.K., D.G.B, T.L.G, M.O.B. and S.V.B. provided technical expertize and scientific feedback. S.Y.C.Y., L.L.S. and T.J.P. wrote and edited the manuscript.
