## [Peer Review File · Nature Communications]

Pan-cancer analysis of longitudinal metastatic tumors reveals genomic alterations and immune landscape dynamics associated with pembrolizumab sensitivityReviewers' Comments:

Reviewer #1:

Remarks to the Author:

In this article, Yang et al. present an extensive exploration of the molecular determinants of response to a single anti PD1 antibody (pembrolizumab) from a phase II study in metastatic cancer patients of multiple types.

Overall, the study design is remarkable for its completeness, gathering genomic analysis of the tumor, transcriptomic analysis, immune cell population profiling and ctDNA kinetics across the course of the disease. It is also interesting since it is designed to find molecular traits of resistance across many diverse cancer types.

Some of the findings of the phase II have been already presented in 2 articles, both in press currently. One dealing with the ctDNA variation between baseline and the cycle 3 of the treatment as a strong predictor of PFS and OS. The other one presents the association of the germline heterozygous HLA-C locus with reduced clinical benefit.

In the first chapter, the authors present the additional ctDNA analysis in relation to tumor burden variation and also to tumor mutational burden. The definition of the 4 groups is interesting as it shows how ctDNA analysis can complement tumor imaging analysis. The relation to tumor mutational burden is, however, more difficult for me. I do not get really the point of this analysis since it mostly contains descriptive data, not adjusted to the number of patients within each tumor subtypes (ex. HS group contains a large fraction of metastatic melanoma tumors and therefore, knowing that melanoma is the tumor presenting the highest number of mutation, this fact should be discussed). Also, the result that ctDNA variation adds prognostic value to TMB groups is somehow expected since in the previous paper the authors show that ctDNA is an independent predictor of prognostic in a multivariate analysis adjusted to TMB.

The genomic analysis of the tumor at baseline suggests BRCA2 alterations as a determinant of clinical benefit, confirming, however, a previous study on melanoma patients treated by antiPD1. Still, it would be interesting to know if the BRCA2 mutations occurs more frequently in a special type of tumor or not. The statement on line 74-75 could be better supported by the description of the PDL1 status in the patients with clinical benefit (since only descriptive data are shown).

The other genomic alterations investigated are copy number variations and B2M LOH. Here, the results reported only confirm previous reports.

The transcriptional analysis is more interesting: contrary to ctDNA variations that are relevant to assess tumor response, the immune signatures evolution are not correlated to tumor outcome. Gene expression profiling reveals new insights like PLA2G2D expression that need to be explained mechanistically and confirmed by larger studies. Still I do not get the strategy of inhibiting a gene that is highly expressed in patients with clinical benefit, despite some suggestions in the Discussion part.

MINOR POINTS:

-the methods of this article should be better described (for example, some info related to ctDNA variation calculation should be reminded, as the condition to consider a positive ctDNA sample should be) even if some info can be find the already published articles, it is not convenient.

-line 130: what is CBR in comparison to CB?

-line 132: typo mistake on "pembrou"

-line 157: explicitly define to what refer HS/CB. Since HS group is defined by both ctDNA and tumor burden variations between baseline and C3 while CB refers to patients in CR, PR, SD longer than 18 weeks.

-line 172/Figure 2b: the p value written on the figure is not the same as the one reported in the text

-line 200: Figure 2d content does not seem to match exactly the description of the text

-the order of the figure panels or the text should be reassessed so that the figure quotation in the text matches the figure presentation (for example, chapter starting line 219)

-line 235: should be referred to figure 3 e and f rather than 3 d and e.

Reviewer #3:

Remarks to the Author:

The authors have submitted a well written, concise report describing high-level data from a prospectively recruited clinical trial of pembrolizumab in patients with melanoma, HNSCC, TNBC, HG serous ovarian cancer, and other rare subtypes. The investigators should be commended for conducting this type of pharmacodynamic study and they amazingly obtained analyzable pre- and on-treatment samples from 60 patients.

The most compelling data, from a clinical standpoint is the identification of cohorts of patients defined based on change in ctDNA and change in imaging parameters. It is not clear whether there is any overlap between this work and that from their recently approved Nature Cancer manuscript, as this is not yet published, but it would be good to know where the findings end with that manuscript and pick up with this one. The authors should

The findings presented from the serial biopsies, while interesting and from high level/quality samples, are not particularly novel. For example, the genetic data (summarized in figure 2a) is confirmatory but not revelatory and the data from figure 3 is completely predictable; namely we anticipate patients who benefit from pembrolizumab to have the described changes.

The use of TMB is a tricky. While it is called based on top tertile, TMB of 10mut/Mb is used in the label for pembrolizumab. It would be interesting to redo the analysis of fig 1G and 1H with the cutoff of 10 mut/MB.

Finally, the data with PLA2G2D is not presented. It states that will be in fig 5D, but there is no figure 5D. I presume this is an oversight, since the authors spend a paragraph in the Discussion about potential next steps with inhibiting this target.

Some minor issues:

1. Suppl Fig 2, the Red MSER should be HS. Please confirm.
2. page 5, line 132 - pembrolizumab is called "pembrou"

Reviewer #4:

Remarks to the Author:

In this interesting study, Dr. Yang and colleagues have analyzed the WES, ctDNA, transcriptomic, immune features of 106 cancer patients who have received pembrolizumab and correlated the findings with patient's clinical benefit. This study has included a large amount of data. Although most findings have been previously reported, these results will be welcomed in the oncology community overall. Given the relatively small number of patients, mixing cancer types, authors should be cautious when interpreting the data to avoid overstatement. Following are my specific comments.

Major comments

1. This is a cohort of mixing cancer types. It is well known that different cancer types may have different cancer biology and underlying mechanism underlying response to IO treatment. It is clearly that the authors are trying to identify the commonalities, which is reasonable. But this should be clearly discussed.
2. TMB and SCNA can be independent biomarkers predicting response according to the reports. Authors in this paper also reported high percent genome CNAs were associated with intrinsic resistance to pembrolizumab, if tumors be stratified by TMB and CNA in this cohort, can the joint TMB/CAN be a better biomarker?
3. Immune score was found to be associated with clinical benefit. How is the correlation of immune

cells, eg CD8 CTL, CD4 effector, Treg, M1, M2 with clinical outcome in this cohort?

4. There are quite some low-end mistakes such as different fonts, different space between lines and paragraphs and quite some typos. No Figure 5c in the paper as well as legend but had figure. No figure 5d as figure but had legend. These made me think that they put these together in a rush or did not put much efforts in it. This is quite disappointing for a manuscript submitted to high-impact journals.

5. Line 158-161: regarding the 35 genes that were more frequently altered in HS/CB. How did you define the mutations? It is well known that the majority of mutations are passengers without any functional consequences even they are in critical cancer genes. This analysis should be based on mutations leading to potential functional changes. Authors should focus on known mutations or at least use some algorithms to predict their functional impact, like CADD or similar. It is not clear whether this is done. This also applies to the 49 genes associated with IO resistance (Line 179-180).

6. The Δ ctDNA is one of the most interesting findings of this study. For example, if this can help to distinguish pseudoprogression vs true progression. This probably deserves some more discussion.

7. However, the data is not included regarding the ctDNA panel and its mutations. The analysis is not clearly described. Maybe it is being published in another paper as authors indicated. However, it will be helpful for reviewers/readers to have access to these data.

8. Based on "Circulating tumor DNA analysis depicts subclonal architecture and genomic evolution of small cell lung cancer" (Nong et al, Nature Communications, 2018), the clonal mutations in ctDNA is better to assess the tumor burden. Since the authors have WES in treatment naïve samples, it will be interesting to see if they focus on clonal mutations only (present in the pre-treatment samples, at least), will that improve the prediction.

9. The authors did a lot of analyses comparing HS/CB vs LS, which are good. But the immunogenomics features should be compared between pre-treatment specimens and post-treatment specimens, which will potentially provide very novel insight on the immunogenomic evolution under IO treatment, a topic that the filed does not have a lot of data for.

10. Many additional analyses can be done regarding the association between genomic features and immune features.

11. Given the small cohorts, some of novel findings should be validated by independent cohorts since there are more and more published datasets with WES, RNA seq and immune profiling data publically available. Following are some examples.

a. Line 56, whether PLA2G2D was is potential therapeutic target and marker of immune resistance.

b. Line 169, BRCA2 mutation vs TMB and IO response. This should be validated by IO trial data and TCGA data.

Minor comments

1. Line 113, please define how to define Δ ctDNA and Δ TM in method or at the first time it appeared in the context.

2. Line 171, there seemed no difference between the TMB values of BRCA2 mutated tumors and wild-type counterparts ($p=0.09$) in Figure 2b, which is different from what you described in the context ($p<0.05$).

3. Line 210-211: "We also examined the frequency of copy number gains in MYC, losses of PTEN and changes in interferon pathway genes (PDCD1, STAT1, JAK1, and JAK2) (Figure 3A)." This reads pretty "sudden". I assume this may be a response to some reviewers from previous submissions. But you may want to have a sentence or two on why you pick these genes.

4. Line 175, if there was no difference in PD-L1 protein expression levels between tumors by BRCA2 mutation status, how can you conclude that "BRCA2 mutation may enrich for pembrolizumab clinical benefit in patients with low or absent tumor PD-L1 protein expression"?

5. Line 223-224, please describe how to define the IM, IFNG and CYT.

6. Figure 2a legend: "A" should be in line with others as lowercase.

7. Line 296, please illustrate the flow-cytometry data in the supplementary materials.

8. Line 380, since most of the immune cell signature differences were not significant in Figure 6b, it is not rational to conclude that "T and B cell transcription programs is associated with anti-PD-1 response".

9. Please adjust the characters in the supplementary figures into the same size.
10. The introduction part is very short and most of the intro was the summary of the findings in this article.
11. More and more cancers are being treated with combination IO or chemoIO, the findings should be discussed in the context.

Responses to Reviewer #1 (Expert in circulating tumor DNA)

Major Points

1. In the first chapter, the authors present the additional ctDNA analysis in relation to tumor burden variation and also to tumor mutational burden. The definition of the 4 groups is interesting as it shows how ctDNA analysis can complement tumor imaging analysis. The relation to tumor mutational burden is, however, more difficult for me. I do not get really the point of this analysis since it mostly contains descriptive data, not adjusted to the number of patients within each tumor subtypes (ex. HS group contains a large fraction of metastatic melanoma tumors and therefore, knowing that melanoma is the tumor presenting the highest number of mutation, this fact should be discussed).

Thank you for the feedback. To clarify our intent to identify molecular (genomic and immune) characteristics associated with pembrolizumab response across cancer types, we have added this description to the introduction to state the rationale to consider our dataset as a whole (and addresses similar comment #1 from Reviewer 4) [Line 70 - 83]:

High tumor mutation burden (high-TMB) has emerged as the most promising and controversial pan-cancer biomarker for predicting ICB therapeutic responses^{3,4}. Despite pan-cancer US FDA approval of ICB treatment for any high-TMB tumor, high-TMB status failed to predict improved ICB response across cancer types in a recent assessment with over 1,500 tumors⁴, calling into question its clinical utility. In recent years, tumor genomics studies enabled by large, multi-dimensional datasets - such as the The Cancer Genome Atlas (TCGA), identified links between genomic alterations in cancers, infiltrating immune cell populations and spontaneous local immune cytolytic activity⁵ to suggest that immune evasion strategies are fundamental to tumor development and may impact ICB response across cancer types. Molecular characteristics uncovered in these studies have potential as predictive biomarkers or novel therapeutic targets to improve ICB clinical benefit. Together, this forms a strong rationale to pursue ICB response biomarker discovery and assessments in pan-cancer cohorts.

We also directly articulated the rationale for the analysis of TMB across our ctDNA-defined comments as follows in the Results [Line 153 - 164]:

To determine the concordance between pembrolizumab sensitivity subgroups and established ICB predictive biomarkers, we evaluated the distributions of TMB and PD-L1 protein expression within the HS subgroup. TMB as a continuous measure was significantly higher in HS tumors compared to the overall cohort (mean 7.72 mut/Mb vs. 1.74 mut/Mb, $p < 0.05$, one-sided Kolmogorov-Smirnov test) (Fig.1F). The majority (14 of 16) of HS TMB fell within the top-tertile of the study distribution, including 6 of 7 tumors with high-TMB ($TMB \geq 10$

mut/Mb). PD-L1 expression was only detected (PD-L1 MPS > 50%) in half (5 of 10) of the HS tumors at baseline with available immunohistochemistry (IHC) data (Supplementary Fig.2). Together, the discordance between TMB, PD-L1 and HS classification, suggest that combined Δ ctDNA and Δ TM subtyping better predicts ICB response in a pan-cancer setting, as compared to stratification by TMB and PD-L1 with pre-specified cutoffs.

While we would like to refine our analysis to account for cancer-type specific effects, we are limited by the sample size of our dataset.

2. Also, the result that ctDNA variation adds prognostic value to TMB groups is somehow expected since in the previous paper the authors show that ctDNA is an independent predictor of prognostic in a multivariate analysis adjusted to TMB.

In the analysis presented, our goal is to explore differences in the prognostic benefit of ctDNA in TMB-high and TMB-low groups. While we have already demonstrated that ctDNA is an independent predictor in a multivariate analysis adjusted to TMB, the previous analyses considered TMB as a continuous variable and lacked statistical significance for OS, PFS, and clinical benefit. In our updated manuscript and in response to Comment 3 from Reviewer 3, we can now show that TMB, when considered as a thresholded variable (TMB > 10), is associated with OS, PFS and CB, consistent with current literature. We are also now able to show that ctDNA dynamics further explain responders and non-responders within high and low TMB groups.

3. The genomic analysis of the tumor at baseline suggests BRCA2 alterations as a determinant of clinical benefit, confirming, however, a previous study on melanoma patients treated by antiPD1. Still, it would be interesting to know if the BRCA2 mutations occurs more frequently in a special type of tumor or not.

In the TCGA dataset of primary tumours, *BRCA2* somatic mutations occur most frequently in Uterine Endometrioid Carcinomas (18%) [cBioportal.org]. Given the limited sample size of each cancer-type cohort in our study and small number of *BRCA2* mutation events (9 of 72), we do not have the statistical power to detect cancer-type associated enrichments. To provide details regarding the distribution of *BRCA2* mutations across our dataset, we added this statement to the results section [Line 190 - 191]:

BRCA2 mutation frequency is 12.5% (9 of 72) across the full study and similarly distributed across cancer cohorts (HGSC: 0/19, TNBC: 1/21, HGSC 2/21, MM: 2/12, Mixed: 4/28).

4. The statement on line 174-175 could be better supported by the description of the PDL1 status in the patients with clinical benefit (since only descriptive data are shown).

A great suggestion that we have now acted upon. Specifically, we included a description of PD-L1 status and clinical benefit rate to better support our finding [Line 206- 209]:

While we observe a higher clinical benefit rate (CBR = 35%, 18 of 51) in PD-L1 expressing tumors (PD-L1 MPS > 0%) compared to tumors without PD-L1 expression (CBR = 15%, 8 of 52), PD-L1 was only expressed in 3 of 9 tumors with *BRCA2* mutations (Fig.2C).

5. The other genomic alterations investigated are copy number variations and B2M LOH. Here, the results reported only confirm previous reports.

We believe there is value in stating confirmation of previous reports, in addition to other novel aspects of our study. Of note, our work also provides new evidence for these events associated with ICB response in cancer types in which it has not been previously reported (One patient of Adenocarcinoma of the GEJ, and two patients of Squamous cell carcinoma of the anus).

6. The transcriptional analysis is more interesting: contrary to ctDNA variations that are relevant to assess tumor response, the immune signatures evolution are not correlated to tumor outcome. Gene expression profiling reveals new insights like *PLA2G2D* expression that need to be explained mechanistically and confirmed by larger studies. Still I do not get the strategy of inhibiting a gene that is highly expressed in patients with clinical benefit, despite some suggestions in the Discussion part.

In our study, we demonstrated that *PLA2G2D* has a gene-expression regulation pattern similar to other known immune checkpoint molecules (i.e. *PDCD1* and *TIGIT*) in response to anti-PD1 blockade suggesting its role in immune regulation within the tumor. In a pan-cancer analysis of TCGA primary tumour gene expression, *PLA2G2D* was identified as a non-CTL/NK gene associated with immune cytolytic activity score (Rooney et al. 2015 Cell). While *PLA2G2D* appears to be involved in tumor immunity, upon further review of current literature, we agree with the Reviewer that proposing therapeutic inhibition on this target is premature at the current moment. We have modified our discussion of *PLA2G2D* to reflect this as follows

In the Results section [Line 359-371]:

Interestingly, we identified increased expression of *PLA2G2D* (Fig.4D), a secreted phospholipase with experimental evidence to suggest its function to attenuate T helper 1 immune responses by maintaining the steady-state levels of anti-inflammatory lipids in murine lung tissues during viral infections²⁹. In a pan-cancer analysis with TCGA data of primary tumors⁵, *PLA2G2D* expression was correlated with CYT, to a similar degree with other immunosuppressive factors and interferon-stimulated T-cell attracting cytokines. We further validated its upregulation in an independent dataset of 62 metastatic melanoma tumors upon nivolumab treatment²⁰ ($p = 0.078$ two-sided Wilcoxon rank sum test). We also found evidence that responders to anti-PD1 or anti-PDL1 had higher *PLA2G2D* expression in pre-therapy tumors compared to non-responders in urothelial cancer³⁰ and anti-CTLA4 pre-treated advanced melanoma²⁰ data sets ($p < 0.10$,

uncorrected for multiple testing, Supplementary Fig.11)31. Together, this data suggest the potential of *PLA2G2D* as a biomarker of tumor immunity.

In the Discussion section [Line 464- 474]:

We identified and validated increased phospholipase *PLA2G2D* expression in tumors upon anti-PD-1 treatment. While the bulk of current literature has characterized the role of Pla2g2d in fatty-acid metabolism within the contexts of autoimmune disorders and viral infections^{41,42,43}, little is known of its relevance in *cancer* development and immunotherapy response. Rooney et al.⁵ reported that *PLA2G2D* expression is correlated with the cytolytic activity score to suggest its participation in counter-regulatory activities that limit immune responses in primary solid tumors⁵. Moreover, Miki et al.⁴⁴, in a melanoma mouse model, demonstrated that Pla2g2d deficiency is associated with delayed tumor growth. Our findings along with growing literature, together suggest that future investigations to elucidate the mechanistic role of *PLA2G2D* in tumor contexts will enrich our understanding of tumor immunobiology.

Minor Points

1. The methods of this article should be better described (for example, some info related to ctDNA variation calculation should be reminded, as the condition to consider a positive ctDNA sample should be) even if some info can be find the already published articles, it is not convenient.

We have now provided the following details for quantifying ctDNA levels in the ctDNA methods section [Line 531 - 537]:

For each baseline and on-treatment time-point plasma sample, absolute ctDNA levels were quantified as the mean variant allele frequencies after normalization by plasma volumes collected across all 16 variants tested in units of mutant molecules per mL of plasma used for extraction (MTM per mL). The early change in ctDNA was calculated as the percentage difference in absolute ctDNA levels between cycle3 and baseline time-points. An increase was defined greater than zero, while decrease was less than zero.

2. line 130: what is CBR in comparison to CB?

Thank you for identifying this inconsistency. We have corrected this to “CB” and further clarified the use of CBR (clinical benefit rate) where the frequency of CB is reported.

3. line 132: typo mistake on “pembrou”

We have corrected this to “pembrolizumab”.

4. line 157: explicitly define to what refer HS/CB. Since HS group is defined by both ctDNA and tumor burden variations between baseline and C3 while CB refers to patients in CR, PR, SD longer than 18 weeks.

We modified this line as follows to define the HS/CB group [Line 181 - 184]:

To uncover genomic alterations associated with pembrolizumab response, we identified genes that were more frequently altered by non-synonymous germline or somatic mutations in **patients who experienced CB or high pembrolizumab sensitivity** (HS/CB; n = 19) compared to patients with low sensitivity (LS; n = 27).

5. line 172/Figure 2b: the p value written on the figure is not the same as the one reported in the text

Thank you for spotting this inconsistency. We updated the in-text results to reflect two-sided test results reported in Figure 2b as follows [Line 201-203]:

We observed that tumors with germline and/or somatic BRCA2 mutation have significantly higher TMB compared to tumors without mutated BRCA2 ($p < 0.10$, Wilcoxon rank sum test, two-sided) (Fig.2B), consistent with previous studies⁹.

6. line 200: Figure 2d content does not seem to match exactly the description of the text

We made the following in-text corrections (Line 243 - 245):

The frequency of *B2M* LOH was higher in LS compared to HS/CB tumors (12/24, 50% vs 2/16, 13%) (Fig.2A). *B2M* LOH was not detected in HS tumors (0%, 0/9) (Fig.2D).

7. the order of the figure panels or the text should be reassessed so that the figure quotation in the text matches the figure presentation (for example, chapter starting line 219)

We reviewed and updated all figure numbers in the manuscript to ensure they are referring to the correct figure panels that are presented.

8. line 235: should be referred to figure 3 e and f rather than 3 d and e.

We have addressed this in minor point #7.

Responses to Reviewer #3 (Expert in biomarkers/immunotherapies)

Major Points

1. The most compelling data, from a clinical standpoint is the identification of cohorts of patients defined based on change in ctDNA and change in imaging parameters. It is not clear whether there is any overlap between this work and that from their recently approved Nature Cancer manuscript, as this is not yet published, but it would be good to know where the findings end with that manuscript and pick up with this one. The authors should...[sic]

Thank you for the comment. We updated the citations to include the full citation of our published work in Nature Cancer within the manuscript to provide the background context of our initial ctDNA work.

2. The findings presented from the serial biopsies, while interesting and from high level/quality samples, are not particularly novel. For example, the genetic data (summarized in figure 2a) is confirmatory but not revelatory and the data from figure 3 is completely predictable; namely we anticipate patients who benefit from pembrolizumab to have the described changes.

While we recognize the findings we presented in Figures 2 and 3 have been reported by others, mainly in melanoma and lung cancer, our study highlights the cancer-type agnostic nature of these molecular features (*BRCA2* mutations, *B2M* loss of heterozygosity, and immune infiltration and interferon gamma gene expression signatures) to predict pembrolizumab treatment outcomes in patients with metastatic solid cancers treated in a single trial. At the time of manuscript submission, we were unaware of any existing report of an association *BRCA2* mutation and sensitivity to pembrolizumab in a pan-cancer patient cohort. This finding was recently confirmed by Samstein and colleagues (Nature Cancer, in press) in a retrospective analysis of the MSK-IMPACT pan-cancer cohort of patients treated with immune checkpoint inhibitors (anti-PD1 and or anti-CTLA4). We incorporated this new citation into the discussion [Line 424 - 437]:

In cancer studies, *BRCA2* is often investigated in conjunction with *BRCA1* for their DNA repair tumor suppressor function and association with breast and ovarian cancer risks³⁷. Our results demonstrate that non-synonymous mutations in *BRCA2*, but not *BRCA1*, were more frequently observed in patients with clinical benefit and early molecular sensitivity. In a recent retrospective analysis of the pan-cancer MSKCC-IMPACT patient cohort, Samstein et al.³⁸ identified clinical benefit in 44% of patients with *BRCA2*-deficient cancers following ICB treatment. Our finding, not only supports previous reports of improved response to single-agent ICB in *BRCA2*-deficient tumors, but also emphasizes the need to evaluate *BRCA1* and *BRCA2* mutated patients separately in future clinical trial designs. Due to the limited sample size, subset analysis of germline or somatic SNVs was not explored. Further studies are warranted to examine the functional effect of specific germline *BRCA2* mutations on the activity of immune cells as well as the impact of *BRCA2* mutations on the immunogenicity of tumor cells.

3. The use of TMB is a tricky. While it is called based on top tertile, TMB of 10mut/Mb is used in the label for pembrolizumab. It would be interesting to redo the analysis of fig 1G and 1H with the cutoff of 10 mut/MB.

Thank you for the helpful suggestion, which we have acted upon and included in the Results section as follows [Line 166-177]:

Given that TMB-high (TMB \geq 10 mut/Mb) status is a US FDA-approved pan-cancer biomarker used to select metastatic cancers for ICB therapy, we explored whether Δ ctDNA provides added benefit for risk stratification within TMB-high and TMB-low tumors (TMB < 10 mut/Mb). We compared the OS and PFS of TMB-high and TMB-low groups further divided based on Δ ctDNA (Fig1.G,H). As expected compared to the patients with the least favorable responses (TMB-low and Δ ctDNA > 0, n = 33), we observed the most favorable OS and PFS probabilities in TMB-high and Δ ctDNA < 0 (OS HR = 0.32, p = 0.007; PFS HR = 0.41, p = 0.008, Cox proportional hazards), followed by TMB-low and Δ ctDNA < 0 (n = 18) (OS HR = 0.14, p = 0.05; PFS HR = 0.05, p = 0.005, Cox proportional hazards). These data suggest that Δ ctDNA status provides added value to predict pembrolizumab response within predetermined TMB-high and TMB-low subgroups.

4. Finally, the data with PLA2G2D is not presented. It states that will be in fig 5D, but there is no figure 5D. I presume this is an oversight, since the authors spend a paragraph in the Discussion about potential next steps with inhibiting this target.

Thank you for your careful review. This omission was due to a mis-referencing of the correct figure with the *PLA2G2D* result. We have reviewed and updated all of the figure numbers in the manuscript to ensure they are referring to the correct figure panels that are presented.

Minor Points

1. Suppl Fig 2, the Red MSER should be HS. Please confirm.

Thank you for catching this inconsistency. This has been corrected in supplementary figure 2.

2. page 5, line 132 - pembrolizumab is called "pembrou"

We have corrected this to "pembrolizumab".

Responses to Reviewer #4 (Expert in genomics)

Major Points

1. This is a cohort of mixing cancer types. It is well known that different cancer types may have different cancer biology and underlying mechanism underlying response to IO treatment. It is

clearly that the authors are trying to identify the commonalities, which is reasonable. But this should be clearly discussed.

Thank you for your suggestion. We added a summary of evidence supporting immune evasion mechanisms that are conserved across cancer types to the introduction section (Introduction, Paragraph #2, also listed in our response to Comment #1 from Reviewer #1) [Line 70 - 83]:

High tumor mutation burden (high-TMB) has emerged as the most promising and controversial pan-cancer biomarker for predicting ICB therapeutic responses^{3,4}. Despite pan-cancer US FDA approval of ICB treatment for any high-TMB tumor, high-TMB status failed to predict improved ICB response across cancer types in a recent assessment with over 1,500 tumors⁴, calling into question its clinical utility. In recent years, tumor genomics studies enabled by large, multi-dimensional datasets - such as the The Cancer Genome Atlas (TCGA), identified links between genomic alterations in cancers, infiltrating immune cell populations and spontaneous local immune cytolytic activity⁵ to suggest that immune evasion strategies are fundamental to tumor development and may impact ICB response across cancer types. Molecular characteristics uncovered in these studies have potential as predictive biomarkers or novel therapeutic targets to improve ICB clinical benefit. Together, this forms a strong rationale to pursue ICB response biomarker discovery and assessments in pan-cancer cohorts.

2. TMB and SCNA can be independent biomarkers predicting response according to the reports. Authors in this paper also reported high percent genome CNAs were associated with intrinsic resistance to pembrolizumab, if tumors be stratified by TMB and CNA in this cohort, can the joint TMB/CAN be a better biomarker?

Thank you for the helpful suggestion. Due to space constraints and the need for additional figures to include this analysis, we have now included a new Supplemental Note containing text as followed [Line 966 - 984]:

TMB and PGA as combined biomarker to predict pembrolizumab clinical response

To evaluate tumor genome mutation and copy number burden as a combined biomarker to predict pembrolizumab clinical outcome, we divided the patients into four subgroups, based on the combination of high (>10 mutations/Mb) or low TMB and high (> 50%) or low PGA (Supplementary Fig6). We then calculated the proportion of patients with clinical benefit (CBR) and frequencies of cancer types and pembrolizumab sensitivity types within each subgroup. PGA is low (< 50%) in all six TMB-high tumors (Supplementary Fig6). The TMB-low and PGA-low group contains the fewest tumors with high pembrolizumab sensitivity and more than half of the breast and ovarian cancers. Based on published studies, we expected the highest clinical benefit rate (CBR) in TMB-high and PGA-low tumors and lowest in TMB-low and PGA-high tumors. Indeed, the CBR is 1.9-times higher in the TMB-high and PGA-low group as

compared to the overall CBR (43%, 26/61), while CBR is 3.3-times lower in the TMB-low and PGA-low group (Supplementary Fig6). The proportion of patients with clinical benefit is notably higher in the subgroup of TMB-high and PGA-low (5 out of 6) compared to the subgroup of TMB-low and PGA-high (2 out of 15) patients ($p = 0.006$, Fisher's exact test) (Supplementary Fig6). Our data validated the potential utility of TMB and PGA as a combinatorial biomarker to increase the pool of patients and broaden the range of cancer types that will meaningfully respond to pembrolizumab.

We have included a sentence highlighting this finding in the results section and referred to the more detailed Supplemental Note [Line 234-239]:

When combined with TMB to predict pembrolizumab outcome, we observed that the proportion of patients with CB is notably higher in the subgroup of TMB-high and PGA-low (5 out of 6) compared to the subgroup of TMB-low and PGA-high (2 out of 15) patients ($p = 0.006$, Fisher's exact test) (see detail analysis in supplementary analysis and Supplementary Fig.6).

3. Immune score was found to be associated with clinical benefit. How is the correlation of immune cells, eg CD8 CTL, CD4 effector, Treg, M1, M2 with clinical outcome in this cohort?

An interesting question that we have now incorporated into the Results section as follows [Line 297 - 310]:

When we evaluated the association between cell scores and clinical benefit, PFS and OS, we found that high Tfh was significantly associated with favorable outcomes (log-rank $p < 0.10$, unadjusted for multiple testing, Supplementary Fig.8). Conversely, high neutrophil levels were associated with resistance and diminished overall survival (HR = 0.30, log-rank $p = 0.09$, unadjusted for multiple testing, Supplementary Fig.8). We used the top tertile (greater than 66-percentile) scores of each cell type to define the high signature comparison sample group. Elevated levels of cell types previously observed in the pembrolizumab sensitivity comparison (CD8+ T cells, CD4+ T cell populations including T regulatory cells (Tregs), activated NK cells and M1 macrophages) were notably associated with overall survival (log-rank $p < 0.10$, unadjusted for multiple testing, Supplementary Fig.8). Here, we conclude that measurements of spontaneous immunological activity and infiltrate composition differ between individual tumors and may have potential to predict patient outcome to anti-PD1 treatment.

4. There are quite some low-end mistakes such as different fonts, different space between lines and paragraphs and quite some typos. No Figure 5c in the paper as well as legend but had figure. No figure 5d as figure but had legend. These made me think that they put these together in a rush or

did not put much efforts in it. This is quite disappointing for a manuscript submitted to high-impact journals.

Thank you for the constructive criticism. We have reviewed and corrected the formatting anomalies in the updated submission.

5. Line 158-161: regarding the 35 genes that were more frequently altered in HS/CB. How did you define the mutations?

In the mutation analyses presented, we considered all nonsynonymous somatic events (missense, nonsense, non-stop, translational start site, splice site, in-frame and frame-shift insertions and deletions) and pathogenic germline mutations. We included this description in the methods section [Line 568-570]:

We considered all nonsynonymous somatic events (missense, nonsense, non-stop, translational start site, splice site, in-frame and frame-shift insertions and deletions) and pathogenic germline mutations.

6. It is well known that the majority of mutations are passengers without any functional consequences even they are in critical cancer genes. This analysis should be based on mutations leading to potential functional changes. Authors should focus on known mutations or at least use some algorithms to predict their functional impact, like CADD or similar. It is not clear whether this is done. This also applies to the 49 genes associated with IO resistance (Line 179-180).

We have now performed the suggested analysis and have included the results in the Supplementary analysis section [Line 950-965]:

Enrichment of mutations with predicted functional effects in tumors with high and low pembrolizumab sensitivity

We first used SIFT89 and PolyPhen290 to predict the functional impact of the non-synonymous mutations in each tumor. We selected mutations predicted to be “deleterious” by SIFT and “probably damaging” by PolyPhen2 for the subsequent enrichment analysis. We identified one gene, LRP1B, frequently mutated in the HS/CB group (7 of 19, $p < 0.05$ Fisher’s exact test). In melanoma and non-small cell lung cancer patients, LRP1B mutations have been associated with high TMB and prolonged survival with ICB treatment 91. In our cohort, we observed LRP1B mutations only in melanoma and basal cell carcinoma patients. Due to this association between LRP1B mutations and cancer type, we do not have sufficient evidence to report LRP1B mutations as a pan-cancer biomarker candidate for predicting immune checkpoint blockade response. When we repeated the IO resistance gene analysis, we still did not observe any significant mutation enrichment. We expected this result since we had a limited number of mutations within the genes of interest prior to the functional prediction filter.

7. The Δ ctDNA is one of the most interesting findings of this study. For example, if this can help to distinguish pseudoprogression vs true progression. This probably deserves some more discussion. However, the data is not included regarding the ctDNA panel and its mutations. The analysis is not clearly described. Maybe it is being published in another paper as authors indicated. However, it will be helpful for reviewers/readers to have access to these data. Based on “Circulating tumor DNA analysis depicts subclonal architecture and genomic evolution of small cell lung cancer” (Nong et al, Nature Communications, 2018), the clonal mutations in ctDNA is better to assess the tumor burden. Since the authors have WES in treatment naïve samples, it will be interesting to see if they focus on clonal mutations only (present in the pre-treatment samples, at least), will that improve the prediction.

Thank you for the comment and interest in the ctDNA publication. The full text of the ctDNA report and associated data, including mutations assayed for each patient, is available on-line (<https://www.nature.com/articles/s43018-020-0096-5>). The Natera Signatera bespoke mutation panel design platform for ctDNA selects 16 clonal variants based on whole exome mutation and copy number profiles of patient tumors (<https://clincancerres.aacrjournals.org/content/25/14/4255>).

8. The authors did a lot of analyses comparing HS/CB vs LS, which are good. But the immunogenomics features should be compared between pre-treatment specimens and post-treatment specimens, which will potentially provide very novel insight on the immunogenomic evolution under IO treatment, a topic that the field does not have a lot of data for.

Thank you for the suggestion. We compared the immune-related gene-expression scores (IM, CYT, and IFNG) and CIBERSORT immune cell composition estimates in tumours before and after anti-PD1 treatment and present the findings in a new Results section summarizing the findings of this new analysis as follows [Line 312-324]:

Pembrolizumab therapy induces immune microenvironment sculpting in solid tumors.

We hypothesized that modulations in the immunological activity as a pharmacological consequence of pembrolizumab therapy may be reflected by changes in the transcriptional signatures and immune cell tumor infiltration associated with immune responses. In the subset of 43 patients with available on-treatment tumor gene-expression profiles, we observed that 70% of tumors had increased IMS, IFNG, and CYT scores after pembrolizumab treatment (Fig.3A, Supplementary Fig.10). Notably higher levels of CD8+ T cells, CD4+ memory resting T cells, Tfh, gamma-delta T cells, M1 macrophages, and eosinophils were observed in tumors post- pembrolizumab therapy compared to pre-therapy ($p < 0.10$, unadjusted for multiple testing, two-sided Wilcoxon rank sum test) (Supplementary Fig.10). Our results are consistent with increased immune cell populations in melanoma tumors in response to nivolumab treatment 20. Here, we demonstrate that regardless of tumor type, anti-PD1 therapy acts by

modulating both the immune response and immune cell repertoire within the tumor.

9. Many additional analyses can be done regarding the association between genomic features and immune features.

In response to the all three reviewers, we performed and incorporated several additional analyses to further explore genomic and immune features in our dataset. We anticipate strong interest for this unique dataset as a source for independent validation, meta-analyses, and other exploratory analyses. As such, we have made the sequencing and clinical annotations available through EGA and github repositories. We recently approved multiple data access requests for the data in this paper to support a meta-analyses effort to identify pan-cancer predictive gene-expression-based biomarkers for immune checkpoint therapy. We welcome and look forward to supporting future collaborations and application of this resource by the cancer immunotherapy research community.

10. Given the small cohorts, some of novel findings should be validated by independent cohorts since there are more and more published datasets with WES, RNA seq and immune profiling data publically available. Following are some examples.
 - a. Line 56, whether PLA2G2D was is potential therapeutic target and marker of immune resistance.

We performed the suggested validation using the data available on the TISIDB portal (<http://cis.hku.hk/TISIDB/>) and included the description in the Results and a new Supplemental Figure as follows [Line 365-370]:

We further validated its upregulation in an independent dataset of 62 metastatic melanoma tumors upon nivolumab treatment 20 ($p = 0.078$ two-sided Wilcoxon rank sum test). We also found evidence that responders to anti-PD1 or anti-PDL1 had higher PLA2G2D expression in pre-therapy tumors compared to non-responders in urothelial cancer³⁰ and anti-CTLA4 pre-treated advanced melanoma²⁰ datasets ($p < 0.10$, uncorrected for multiple testing, Supplementary Fig.11)³¹.

- b. Line 169, BRCA2 mutation vs TMB and IO response. This should be validated by IO trial data and TCGA data.

We performed the suggested validation using independent cohorts and included the description in the Results and a new Supplemental Figure as follows [Line 199-206]:

We further investigated the relationship of *BRCA2* mutation status with known biomarkers of response: TMB and PD-L1 protein expression. We observed that tumors with germline and/or somatic *BRCA2* mutation have significantly higher TMB compared to tumors without mutated *BRCA2* ($p < 0.10$, Wilcoxon rank sum test, two-sided) (Fig.2B), consistent with previous studies⁹. We further validated this significant association using publicly available mutation and TMB data from

three independent publicly available pan-cancer data sets (Broad MSS mixed solid tumors¹⁰, UMich MET50011, and MSKCC-IMPACT IO study¹²) (Supplementary Fig.3).

Minor Points

1. Line 113, please define how to define Δ ctDNA and Δ TM in method or at the first time it appeared in the context.

We have added clarifying definitions to Results [Line 131-134]:

When assessing the correlation between the change of ctDNA (Δ ctDNA) and the change of target lesion measurement (Δ TM) between baseline and 6-7 weeks of treatment, we observed that, together, these metrics stratified patients into four subgroups associated with distinct survival outcomes (Fig.1C-E).

2. Line 171, there seemed no difference between the TMB values of BRCA2 mutated tumors and wild-type counterparts ($p=0.09$) in Figure 2b, which is different from what you described in the context ($p<0.05$).

Thank you for spotting this inconsistency. We addressed this previously for Reviewer 1, Comment 5. We have updated the in-text results to reflect two-sided test results reported in Figure 2b as follows [Line 200-203]:

We observed that tumors with *BRCA2* mutation (germline and somatic) have significantly higher TMB compared to tumors without mutated *BRCA2* ($p < 0.10$, Wilcoxon rank sum test, two-sided) (Fig.2B), consistent with previous studies⁹.

3. Line 210-211: “We also examined the frequency of copy number gains in *MYC*, losses of *PTEN* and changes in interferon pathway genes (*PDCD1*, *STAT1*, *JAK1*, and *JAK2*) (Figure 3A).” This reads pretty “sudden”. I assume this may be a response to some reviewers from previous submissions. But you may want to have a sentence or two on why you pick these genes.

We modified this sentence in the Results section to provide rationale and supporting references for selecting the particular variants [Line 255-258]:

Lastly, we examined the frequency of somatic copy number alterations (SCNA) associated with regulation of anti-tumor immune responses in pre-clinical and clinical studies: gains in *MYC* 16, losses of *PTEN*¹⁷ and gains or losses in interferon pathway genes (*PDCD1*, *STAT1*, *JAK1*, and *JAK2*) (Fig.2A).

4. Line 175, if there was no difference in PD-L1 protein expression levels between tumors by BRCA2 mutation status, how can you conclude that “BRCA2 mutation may enrich for pembrolizumab clinical benefit in patients with low or absent tumor PD-L1 protein expression”?

In hindsight, we agree the statement is confusing and have modified the section for clarity as follows. The changes are shown below [Line199-211]:

We further investigated the relationship of *BRCA2* mutation status with known biomarkers of response: TMB and PD-L1 protein expression. We observed that tumors with germline and/or somatic *BRCA2* mutation have significantly higher TMB compared to tumors without mutated *BRCA2* ($p < 0.10$, Wilcoxon rank sum test, two-sided) (Fig.2B), consistent with previous studies⁹. We further validated this significant association using publicly available mutation and TMB data from three independent publicly available pan-cancer data sets (Broad MSS mixed solid tumors¹⁰, UMich MET50011, and MSKCC-IMPACT IO study¹²) (Supplementary Fig.3). While we observe a higher clinical benefit rate (CBR = 35%, 18 of 51) in PD-L1 expressing tumors (PD-L1 MPS > 0%) compared to tumors without PD-L1 expression (CBR = 15%, 8 of 52), PD-L1 was only expressed (MPS > 0%) in 3 of 9 tumors with *BRCA2* mutations (Fig.2C). This finding suggests that *BRCA2* mutation status may predict pembrolizumab response, independent of PD-L1 expression.

5. Line 223-224, please describe how to define the IM, IFNG and CYT.

Please refer to the methods and associated citations associated with each published signature [Line 270-274]:

We investigated previously published gene-expression signatures for total immune infiltration (IM; inferred total immune infiltration score)¹⁸, interferon-gamma signaling (IFNG)¹⁹, cytolytic activity (CYT)⁴ and abundances of 22 immune cell subpopulations inferred by CIBERSORT deconvolution analysis in 65 baseline tumors (CB rate = 30%).

6. Figure 2a legend: “A” should be in line with others as lowercase.

Thank you for your careful review. This has been re-formatted in the final version.

7. Line 296, please illustrate the flow-cytometry data in the supplementary materials.

We have now added a figure illustrating our flow-cytometry data gating strategy to the supplementary materials (Supplemental Fig9A).

8. Line 380, since most of the immune cell signature differences were not significant in Figure 6b, it is not rational to conclude that “T and B cell transcription programs is associated with anti-PD-1 response”.

We have modified this subtitle to “*Differential regulation of T and B cell activation and signaling transcription programs in anti-PD-1 treated tumors is predictive of clinical response*”.

9. Please adjust the characters in the supplementary figures into the same size.

Thank you for the suggestion. We reviewed and re-formatted all supplementary figures.

10. The introduction part is very short and most of the intro was the summary of the findings in this article.

Thank you for the suggestion. In response to major point #1, we expanded the introduction to include a summary of evidence supporting immune evasion mechanisms that are conserved across cancer types.

11. More and more cancers are being treated with combination IO or chemoIO, the findings should be discussed in the context.

We have added the following statements to the Discussion Section [Line 492-496]:

An increasing number of clinical trials are being performed evaluating ICB in combinations with chemotherapy, molecularly targeted therapy, and other immunotherapeutic agents (Upadhaya et al). As such, delineation of the molecular and immune milieu before and after ICB monotherapy is crucial to provide a benchmark against which the effects of additional antitumor agents can be investigated.

Reviewers' Comments:

Reviewer #1:

Remarks to the Author:

The majority of the points have been addressed by the authors.

However, I don't really get the point line 155-162 that the data demonstrate a discordance between TMB status and sensitivity to pembrolizumab. For me the data are concordant with a high TMB and a good response to the therapy. It therefore only appears that PDL1 expression is discordant. The authors could have provided survival analysis based on TMB or PDL1 expression to strengthen their argumentation.

Line 359: the authors do not describe the figure in details. It appears that there is an increase in both HS/CB and LS groups, even if the increase extent is higher in the CB/HS group. This would deserve to be more clearly written.

Reviewer #3:

Remarks to the Author:

The authors have done a nice job of responding to each of the reviewers critiques, and have submitted a more compelling manuscript. I do not have any further, specific comments.

Reviewer #4:

Remarks to the Author:

The authors have done extensive work in the revised manuscript. My comments are satisfactorily addressed.

Responses to Reviewer 1 comments:

1. The majority of the points have been addressed by the authors. However, I don't really get the point line 155-162 that the data demonstrate a discordance between TMB status and sensitivity to pembrolizumab. For me the data are concordant with a high TMB and a good response to the therapy. It therefore only appears that PDL1 expression is discordant. The authors could have provided survival analysis based on TMB or PDL1 expression to strengthen their argumentation.

Thank you for the comment. Upon reviewing, we agree that the message of the section was unclear and have made changes to clarify the intent and interpretation of our analysis. Shown below is the full section for context and the modifications highlighted:

To determine the concordance between pembrolizumab sensitivity subgroups and established ICB predictive biomarkers based on pre-specified universal cutoffs, we evaluated the distributions of TMB and PD-L1 protein expression within the HS subgroup. TMB as a continuous measure was significantly higher in HS tumors compared to the overall cohort (mean 7.72 mut/Mb vs. 1.74 mut/Mb, $p < 0.05$, one-sided Kolmogorov-Smirnov test) (Fig.1F). The majority (14 of 16) of HS TMB fell within the top-tertile of the study distribution, of which only 6 tumors (6 of 16; 38%) met the TMB-high biomarker criteria ($TMB \geq 10$ mut/Mb). PD-L1 expression was only detected (PD-L1 MPS > 50%) in half (5 of 10) of the HS tumors at baseline with available immunohistochemistry (IHC) data (Supplementary Fig.2). Together, the discordance between TMB-high, PD-L1 and HS classification, suggest that treatment opportunities in otherwise treatment-sensitive patients would be lost when using current pre-specified cutoffs of TMB and PD-L1 scoring to select patients for anti-PD1 treatment in a pan-cancer setting.

2. Line 359: the authors do not describe the figure in details. It appears that there is an increase in both HS/CB and LS groups, even if the increase extent is higher in the CB/HS group. This would deserve to be more clearly written.

Thank you for the suggestion. We have modified the text as follows:

Interestingly, in both LS and HS/CB samples we identified increased expression of PLA2G2D (Fig.4D), with a trend towards greater magnitude of increase in the HS/CB group.